



# Estimating Early-Winter Antarctic Sea Ice Thickness From Deformed Ice Morphology

M. Jeffrey Mei[1,2], Ted Maksym[1], and Hanumant Singh[3]

[1]Department of Applied Ocean Science and Engineering, Woods Hole Oceanographic Institution, MA 02540, United States
[2]Department of Mechanical Engineering, Massachusetts Institute of Technology, MA 02139, United States
[3]Department of Electrical and Computer Engineering, Northeastern University, MA 02115, United States

*Correspondence to:* Jeffrey Mei (mjmei@mit.edu)

**Abstract.** Satellites have documented variability in sea ice areal extent for decades, but there are significant challenges in obtaining analogous measurements for sea ice thickness data in the Antarctic, primarily due to difficulties in estimating snow cover on sea ice. Sea ice thickness can be estimated from surface elevation measurements, such as those from airborne/satellite LiDAR, by assuming some snow depth distribution or empirically fitting with limited data from drilled transects from various field studies. Current estimates for large-scale Antarctic sea ice thickness have errors as high as ∼50%, and simple statistical models of small-scale mean thickness have similarly high errors. Averaging measurements over hundreds of meters can improve the model fits to existing data, though these results do not necessarily generalize to other floes. At present, we do not have algorithms that accurately estimate sea ice thickness at high resolutions. We use a convolutional neural network with laser altimetry profiles of sea ice surfaces at 0.2 m resolution to show that it is possible to estimate sea ice thickness at 20 m resolution with better accuracy and generalization than current methods (mean relative errors ∼15%). Moreover, the neural network does not require specifying snow depth/density, which increases its potential applications to other LiDAR datasets. The learned features appear to correspond to basic morphological features, and these features appear to be common to other floes with the same climatology. This suggests that there is a relationship between the surface morphology and the ice thickness. The model has a mean relative error of 20% when applied to a new floe from the region and season, which is much lower than the mean relative error for a linear fit (errors up to 47%). This method may be extended to lower-resolution, larger-footprint data such as such as IceBridge, and suggests a possible avenue to reduce errors in satellite estimates of Antarctic sea ice thickness from ICESat-2 over current methods, especially at smaller scale.

## 1 Introduction

Satellites have documented changes in sea ice extent (SIE) for decades (Parkinson and Cavalieri, 2012); however, sea ice thickness (SIT) is much harder to measure remotely. Declines in Arctic sea ice thickness over the past several decades have been detected in under-ice upward-looking sonar surveys and satellite observations (Rothrock et al., 2008; Kwok and Rothrock, 2009). Arctic ice thickness has been observed with satellite altimetry to continue to decline over the past decade (Kwok and Cunningham, 2015), but any possible trends in Antarctic SIT are difficult to detect because of the presumably relatively small changes, and difficulties in estimating sea ice thickness in the Antarctic (Kurtz and Markus, 2012; Zwally et al., 2008). Because



fully-coupled models generally fail to reproduce the observed multi-decadal increase in Antarctic SIE, it is likely that their simulated decrease in Antarctic SIT is also incorrect (Turner et al., 2013; Shu et al., 2015). However, ocean-ice models forced with atmospheric reanalysis correctly reproduce an increasing Antarctic SIE and suggest an increasing SIT (Holland et al., 2014). Massonnet et al. (2013) found that assimilating sea ice models with sea ice concentration shows that SIT covaries

positively with SIE at the multi-decadal time scale, and thus implies an increasing sea ice volume in the Antarctic. Detection of variations in sea ice thickness and volume are important to understanding a variety of climate feedbacks (e.g. Holland et al., 2006; Stammerjohn et al., 2008); for example, they are critical to understanding trends and variability in Southern Ocean salinity (e.g. Haumann et al., 2016). At present, large-scale ice thickness cannot be retrieved with sufficient accuracy to detect witeh any confidence the relatively small trends in thickness expected (Massonnet et al., 2013), or even interrannual variability

(Kern and Spreen, 2015).

The main source of Antarctic SIT measurements comes from ship-based visual observations (ASPeCt, the Antarctic Sea Ice Processes and Climate program, compiled in Worby et al. (2008)), drill-line measurements (e.g. Tin and Jeffries, 2003; Ozsoy-Cicek et al., 2013), aerial surveys with electromagnetic induction (e.g. Haas et al., 2009) and sporadic data from moored ULS (e.g. Worby et al., 2001; Harms et al., 2001). These are all sparsely conducted, with significant gaps in both time and space,

making it hard to infer any variability or trends. There is also some evidence of a sampling bias towards thinner ice due to logistical constraints of ships traversing areas of thick and deformed ice (Williams et al., 2015).

The only currently-feasible means of obtaining SIT data on a large enough scale to examine thickness variability is through remotely-sensed data, either from large-scale airborne campaigns such as Operation IceBridge (OIB) (Kurtz, 2013), or more broadly from satellite altimetry, (e.g. ICESat (Zwally et al., 2008), or more recently, ICESat-2 (Markus et al., 2017)). Here,

SIT is derived from either the measured surface elevation in the case of laser altimeters (ICESat and OIB), or from a measure of the ice surface freeboard (CryoSat-2) (Wingham et al., 2006). The measurement of the surface elevation itself has some error, due to the error in estimating the local sea surface height (Kurtz et al., 2012). When using radar altimetry, the ice-snow interface may be hard to detect as observations suggest that the radar return can occur from within the snowpack (e.g. Willatt et al., 2009), possibly due to scattering from brine wicked up into the overlying snow, or melt-freeze cycles creating ice lenses,

or from the snow-ice interface (Fons and Kurtz, 2019). However, even with an accurate measurement of the surface elevation, there are challenges with converting this to a SIT estimate.

Assuming hydrostatic equilibrium, the ice thickness $T$ may be related to the surface elevation $F$ (i.e. snow depth + ice freeboard, sometimes called snow freeboard; see Fig. 1) and snow depth $D$ measurements using the relation

$$T = \frac{\rho_w}{\rho_w - \rho_i} F - \frac{\rho_w - \rho_s}{\rho_w - \rho_i} D \qquad (1)$$

for some densities of ice, water and snow $\rho_i, \rho_w, \rho_s$ (Fig. 1). In this article, freeboard refers exclusively to ice freeboard. Without simultaneous snow depth estimates (e.g. from passive microwave radiometry (Markus and Cavalieri, 1998) or from ultrawideband snow radar such as that used on OIB (e.g. Kwok and Maksym, 2014), some assumption of snow depth has to be made, or an empirical fit to field observations is needed (e.g Ozsoy-Cicek et al., 2013). When averaging over multiple kilometers, and in particular during spring, it is common to assume that there is no ice component in the surface elevation, i.e.



$F = D$ in Eq. 1 (Xie et al., 2013; Yi et al., 2011; Kurtz and Markus, 2012). However, this assumption is likely not valid near areas of deformed ice, which may have significant non-zero ice freeboard, and OIB data suggest this is not true at least for much of the spring sea ice pack (Kwok and Maksym, 2014). More generally, empirical fits of SIT to $F$ can be used (Ozsoy-Cicek et al., 2013), but these implicitly assume a constant proportion of snow within the surface elevation and a constant snow and ice

density. These are not likely to be true, particularly at smaller scales and for deformed ice. Moreover, detecting variability with such methods is prone to error because these relationships may change seasonally and interannually. More recently, Li et al. (2018) has used a regionally- and temporally-varying density (equivalently, a variable proportion of snow in surface elevation) inferred from the empirical fits of Ozsoy-Cicek et al. (2013) and Xie et al. (2011), which is equivalent to a more complex, regime-dependent set of snow assumptions.

A key question is how much the sea ice morphology affects these relationships between surface measurements and thickness. Pressure ridges, which form when sea ice collides, fractures and forms a mound-like structure (Fig. 1), are a primary source of deformed ice. Although only a minority of the sea ice surface is deformed, ridges occur at a spatial frequency of 3-30 per km and so may account for a majority of the total sea ice volume (Worby et al., 1996; Haas et al., 1999). Around such areas, both the ice freeboard and snow depth may be high, and we do not yet understand the distribution of snow around such deformation

features. This means that local estimates of SIT are likely biased low as the average ice freeboard cannot be assumed to be zero. Moreover, the effective density of deformed ice (i.e. the density of the deformed ice including snow-, air- and seawater-filled gaps) may differ significantly from level ice areas due to drained brine and trapped snow in ridge sails, and seawater in large pore spaces in ridge keels (Fig. 1). Because these densities affect the empirical fits, it is important to quantify how SIT predictions should be adjusted to account for morphological differences in surface elevation measurements.

Many pressure ridges can be observed from above using airborne or terrestrial lidar scans (e.g Dierking, 1995). However, it is difficult to derive SIT of deformed areas from these scans due to the difficulty in determining the contribution of snow to the surface elevation measured by a lidar scan. Among other factors, radar-based estimates of snow depth are known to be highly sensitive to surface roughness, weather and grain size (Stroeve et al., 2006; Markus and Cavalieri, 1998). Kern and Spreen (2015) found that snow depth measured by the Advanced Microwave Scanning Radiometer - Earth Observing

System (AMSR-E) around deformed ice is underestimated by a factor of two or more, and found an estimate of Antarctic sea ice thickness error from ICESat of around $50\%$, which is not easily reduced due to AMSR-E snow depths not reporting any uncertainty. Kern and Spreen (2015) also showed that the error estimate in the sea ice thickness is considerably affected by the snow depth error, with a conservative estimate of 30% error in snow depth leading to a relative ice thickness error up to $80\%$.

Sea ice draft and ridge morphology may also be observed from below using sonar on autonomous underwater vehicles

(AUVs) (e.g Williams et al., 2015). As most of the sea ice is below water, using the mean draft as a direct estimate of the SIT gives lower errors than surface elevation-based methods. Moreover, the underside of the deformed ice surface does not have snow, making the morphological features less obscured. Although AUV datasets of deformed ice have higher resolution than air- and satellite-borne lidar datasets, they are much more sparsely conducted and fewer such datasets exist. This makes it hard to generalize conclusions of deformed sea ice from empirical datasets. It is therefore important to understand how the

morphology of deformed ice relates to its thickness distribution. By using coincident, high-resolution and three-dimensional





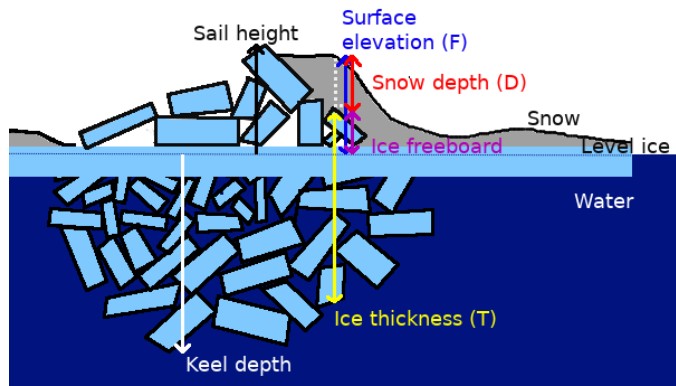

**Figure 1.** A schematic diagram of a typical first-year ridge. The ridge may not be symmetric, and peaks of the sail and keel may not coincide. The effective density of the ice is affected by the air gaps above water and the water gaps below water. $T$, $D$ and $F$ may be linked by assuming hydrostatic balance (Eq. 1).

AUV and lidar surveys of deformed ice, we can characterize areas of deformation and surface morphology and its relationship to ice thickness and surface elevation much better than with linear, low-resolution drilling profiles.

In order to account for the varying effective density of a ridge, we need to be able to characterize different deformed surfaces. The analysis of ridge morphology is currently very simplistic. As summarized in Strub-Klein and Sudom (2012),
the geometry of the above-water (sail) and below-water (keel) heights is typically analyzed, traditionally by calculating the sail-keel ratios and sail angles (Timco and Burden, 1997). There are known morphological differences between Arctic and Antarctic ridges, such as sail heights of Antarctic ridges being generally lower than those of Arctic ridges, but these are not known comprehensively (Tin and Jeffries, 2003). According to drilling data and shipboard underway observations, Antarctic ridges have typical sail heights of less than 1 m (Worby et al., 2008) and keel depths of order 2-4 m (Tin and Jeffries, 2003),
though much thicker (maximum keel depths > 15 m) ridges have also been observed with AUVs (Williams et al., 2015). Metrics like sail/keel angle are less meaningful in the presence of non-triangular, irregular or highly deformed ridges (e.g. Fig. 2), which are underrepresented in literature due to selection bias. Arctic ridges are somewhat more well-studied, with Tucker III and Govoni (1981) finding a square-root relationship between block size and above-water (sail) height, and Timco and Burden (1997) finding a linear relationship between sail height and keel depth but no relationship between sail height and
level ice thickness. Ekeberg et al. (2015) found that first-year (Arctic) ridge keels are better characterized by a trapezoid than a triangle, and Petty et al. (2016) found that ice thickness could be predicted (with considerable error) from metrics taken from lidar-derived topography of deformed ice. These results may or may not hold for Antarctic ridges. For Antarctic ridges, Tin and Jeffries (2003) found the keel depth was proportional to the level ice thickness around a ridge, and Tin and Jeffries (2001b) found a linear relationship between the ice thickness and snow surface roughness. It is possible that other, more complex
metrics may be more relevant for characterizing the relationship between pressure ridge morphology and its corresponding SIT distribution. Identifying how the morphology of deformed ice can inform estimates of sea ice thickness is important for reduce





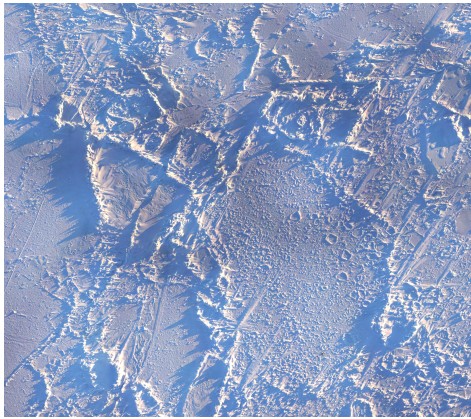

**Figure 2.** Drone imagery (180 m x 180 m) of heavily deformed ice in the Ross Sea, Antarctica. There are multiple ridges which cannot be easily separated. The ridge widths and slopes are varying and must be arbitrarily defined, leading to a variety of possible values. Image provided by Guy Williams.

errors on SIT estimates, which is necessray to understanding temporal-spatial variations in SIT using existing measurements of surface elevation.

The uncertainty in sea ice density is also a contributing factor to the high uncertainty of sea ice thickness estimates (Kern and Spreen, 2015). For example, if assuming zero ice freeboard ($F = D$ in Eq. 1) with some known snow density, a 10%

uncertainty in the sea ice density can lead to a 50% uncertainty in the sea ice thickness. As mentioned before, the effective density may also vary locally. On previous Antarctic fieldwork such as SIPEX-II in spring 2012, Hutchings et al. (2015) found the density of first-year ice in the presence of porous granular ice to be as low as 800 kg m$^{-3}$, a difference of more than 10% from the standard assumption of 900-920 kg m$^{-3}$ (e.g Worby et al., 2008; Xie et al., 2013; Maksym and Markus, 2008; Zwally et al., 2008; Timco and Weeks, 2010), but in line with the 750-900 kg m$^{-3}$ range found by Urabe and Inoue (1988).

This effective density could vary regionally and seasonally in line with ridging frequency, and knowing these variations with greater certainty would decrease the errors in sea ice thickness estimations. The effective density may also vary locally around areas of deformed ice, which have varying gap volumes. This means that the scatter in any given linear fit of $T$ and $F$, and the variability between different fits for different datasets, can be interpreted as differences in effective densities; alternatively, this points out that linear fits will have an irreducible error due to local effective density variations.

In this paper, we aim to use a high-resolution dataset of deformed sea ice to develop better algorithms to estimate sea ice thickness from surface topography. Unlike previous studies which have relied on low-resolution, 2D drilling transects, we use high-resolution, 3D characterization of the snow surface from terrestrial lidar, coincident with 3D ice draft from an autonomous underwater vehicle and detailed snow depth measurements. In particular, having 3D coverage allows for the analysis of complex morphological features. We first analyze our dataset to examine the morphology of the surveyed first-year ridges and potential

relationships with ice thickness. Second, we examine simple statistical relationships between surface elevation and ice thickness and simple measures of local morphology, and compare with prior studies. We also estimate effective densities of ice and snow



and compare with field data. Lastly, we use a deep learning convolutional neural network to improve estimates of local ice thickness by using complex, non-linear functions of 3D surface morphology. We then discuss how learned features in the neural network may be related to physically-meaningful morphological features, and consider possible extensions to this work on larger datasets.

## 2  Data

The PIPERS (Polynas, Ice Production, and seasonal Evolution in the Ross Sea) expedition took place from early April to early June 2017 (Fig. 3). In total, 6 AUV ice draft surveys were taken of the undersides of deformed sea ice. Of these, 4 coincided with snow depth measurements and a lidar survey of the surface elevation, thus providing a 'layer-cake' of snow depth, ice freeboard and ice draft data (following Williams et al. (2013)). These 4 layer cakes are shown in Fig. 4. There are two other

AUV scans which lack lidar/snow measurements but are included for draft-related analysis. The AUV scans were done using the Seabed-class AUV from the Woods Hole Oceanographic Institution equipped with a swath multibeam sonar (Imagenex 837 DeltaT). The AUV multibeam data was processed to correct for vehicle pose, then individual swaths were stitched together, with manual corrections to pitch and roll offsets of the sensors to minimize differences in drafts for overlapping portions of adjacent swaths. This largely follows the methodology in Williams et al. (2015), although Simultaneous Localization and

Mapping (SLAM) algorithms were not applied here as the quality of the multibeam maps were determined to be comparable to those without SLAM processing, and any improvements in resolving small-scale features would not affect the analysis here. The vertical error for is estimated at 10 cm over deformed areas and <1 cm for level areas (Williams et al., 2015). The scans were ultimately binned at 0.2 m horizontal resolution. The surface elevation scans were done with a Riegl VZ-400 lidar, using four scans from different sides of a 100 m x 100 m grid to minimize shadows, which were stitched together

using tripod-mounted targets placed around the grid. The output point cloud was binned at 0.2 m resolution, and any small shadows were interpolated over with natural neighbor interpolation (Sibson, 1981). The snow depth measurements were done using a MagnaProbe, a commercial product by Snow-Hydro LLC with negligible vertical error when measuring snow depth on top of ice (Sturm and Holmgren, 2018; Eicken and Salganek, 2010). The probe was fitted with an Emlid Reach Real-Time Kinematic GPS, referenced to base stations on the floe, which allowed for more precise localization of snow depth. Using Post-

Processed Kinematic (PPK) techniques with the open-source RTKLIB library and correcting for floe displacement/rotation, the localization accuracy was ~10 cm. A typical survey had ~2000 points, with higher resolution (~10 cm) near areas of changing snow surfaces (near deformed ice) and lower resolution (~5m) over flat, level topography. These measurements were converted into a surface by using natural neighbor interpolation, binned at 20 cm to match the lidar and AUV data. The ice thickness can then be calculated by taking (draft) + (surface elevation) - (snow depth).

In addition to these 6 AUV scans from PIPERS, there were 14 additional AUV scans from other experiments (3 from SIPEX-II, 5 from IceBell and 6 from SeaState) combined for analysis (Williams et al., 2015; Thomson et al., 2018). SIPEX-II took place in East Antarctica in September 2012, IceBell in November 2010 in the Bellinghausen/Weddell seas, and SeaState in October 2015 in the Chukchi/Beaufort seas. Note that SeaState surveyed Arctic ice, but we include it for comparison as

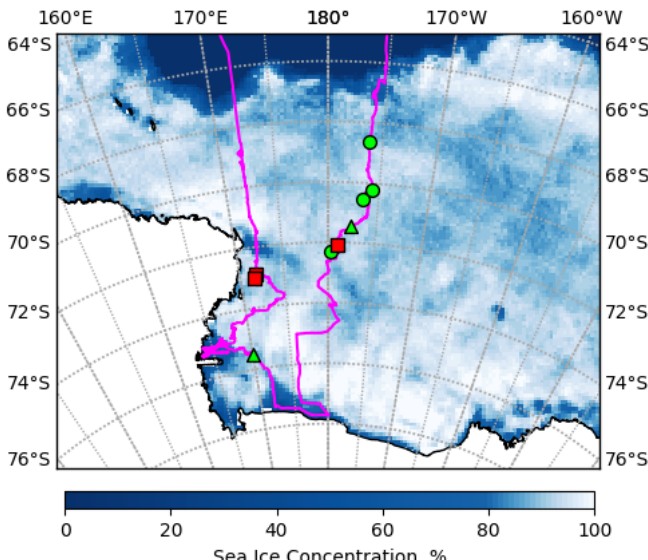

**Figure 3.** PIPERS track (magenta) with locations of ice stations labeled. The AUV scans used in this paper are shown in green (3, 4, 6, 7, 8 and 9) and the other stations (1, 2 and 5) are shown with red squares. Stations 4, 7, 8 and 9 (green circles) also have a surface elevation scan and snow depth measurements; these are shown in Fig. 4. Other stations have some combination of missing lidar/AUV/snow data. Station dates were 05/14 for station 3, 05/24 for station 4, 05/27 for station 6, 05/29 for station 7, 05/31 for station 8 and 06/02 for station 9. Overlain is the sea ice concentration data (5-day median) for 06/02/2017 from ASI-SSMI (Kaleschke et al., 2017).

the AUV scans are primarily of thin, first-year ice. Tin and Jeffries (2003) found that Antarctic ridges are morphologically comparable to those from temperate Arctic ridges which largely form from first-year ice.

## 3 Results

We attempt to statistically model sea ice thickness using surface-measurable metrics (e.g. mean and standard deviation of the
5  surface elevation), in order to see the limitations of this method.

### 3.1 Estimation of Sea Ice Thickness With Surface-Based Metrics

Following previous literature, we expect some relationship between sea ice morphology and thickness. Previous studies have used low-resolution, 2D surveys using drill lines and have found various correlations between certain metrics and the level/deformed ice thickness. The PIPERS surveys comprised floes with ridges that had sails and keels significantly thicker
10  than those that are typically sampled in drilling transects (e.g. Tin and Jeffries, 2003; Worby et al., 2008). For our PIPERS data, we calculate standard metrics such as sail/keel height and angle, summarized in Table 1. The sail/keel angles are not as well-defined for non-linear ridges, so a range of angles is given. The 99th percentile for the sail/keel height is also reported to





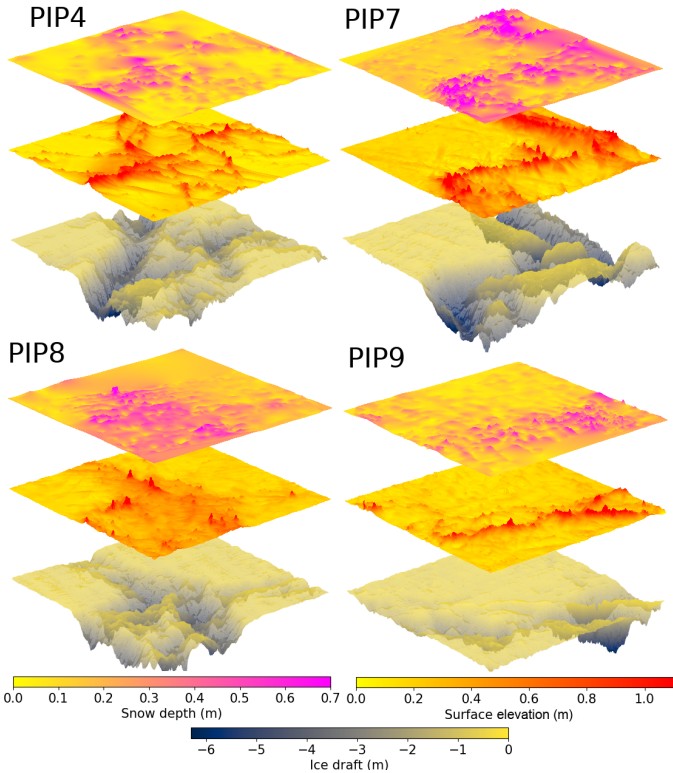

**Figure 4.** Layer cakes from PIPERS. The top layer is the snow depth ($D$), the middle layer is the lidar scan of the surface elevation ($F$), and the bottom layer is the AUV scan of the ice draft. The ice thickness is therefore given by ice draft + surface elevation - snow depth.

**Table 1.** Standard metrics calculated for PIPERS dataset: Sail height ($H_S$), sail angle ($A_S$), the surface roughness (here taken as the standard deviation of the surface elevation, $\sigma$), mean surface elevation ($\bar{F}$), keel depth ($H_K$), keel angle ($A_K$), mean thickness ($\bar{I}$), mean level ice thickness ($\bar{I}_L$), mean deformed ice thickness ($\bar{I}_D$), sail-to-keel ratio ($H_S/H_K$) and % deformation. For $H_S$ and $H_K$, the absolute maximum is given, along with the 99th percentile value of the deformed section draft (in brackets). The amount of deformed ice in each scan is generally high as the survey grids were deliberately chosen for their deformation. The sail/keel angles are not precisely defined because the deformed surfaces are complex and non-linear.

| | $H_S$ (m) | $A_S$ (°) | $\sigma$ (m) | $\bar{F}$ (m) | $H_K$ (m) | $A_K$ (°) | $\bar{I}$ (m) | $\bar{I}_L$ (m) | $\bar{I}_D$ (m) | $H_S/H_K$ | %def. |
|---|---|---|---|---|---|---|---|---|---|---|---|
| PIP4 | 1.64 (1.33) | 6-40 | 0.20 | 0.28 | 7.43 (6.53) | 15-25 | 1.72 | 0.65 | 2.19 | 0.22 (0.20) | 71 |
| PIP7 | 2.02 (1.53) | 3-7 | 0.26 | 0.37 | 7.30 (6.84) | 13-17 | 2.20 | 0.47 | 3.49 | 0.28 (0.22) | 57 |
| PIP8 | 1.95 (1.16) | 1-6 | 0.15 | 0.27 | 5.70 (5.32) | 6-14 | 1.33 | 0.57 | 2.08 | 0.34 (0.22) | 50 |
| PIP9 | 1.82 (1.27) | 6-13 | 0.15 | 0.24 | 6.57 (5.93) | 9-34 | 0.91 | 0.59 | 2.01 | 0.28 (0.21) | 23 |





inhibit the effect of outliers from the lidar/AUV scans. We found the sail/keel ratio was much more consistent when using the 99th percentile values. Our sail angles are typically $< 10^o$ and our keel angles are typically $< 20^o$, in line with averaged values from Tin and Jeffries (2003). However, our sail heights and keel depths are slightly larger in magnitude than the averaged Antarctic values from Tin and Jeffries (2003), and are more similar to their reported values for temperate Arctic ridges. We
compare the relationships for predicting thickness from prior studies that were fitted on low-resolution, 2D datasets with our higher-resolution, 3D dataset to see if the same relationships still hold.

### 3.1.1 Estimating level ice thickness from keel depth

The simplest way to account for the morphology of a deformed surface is to simply measure its maximum height, as has often been done when quantifying ridge statistics from airborne lidar surveys (e.g Dierking, 1995; Petty et al., 2016). A common
way of reporting this is by taking the ratio of the sail height and keel depth. Using the 99th percentile values for the sail/keel from Table 1, the ratio of keel depth and snow-sail height for our PIPERS dataset is 3.9, in line with a ratio of 3.6 from 204 drill profiles of Antarctic sea ice examined by Tin and Jeffries (2001a), and also consistent with a ratio of 4.4 for first-year Arctic ridges from Timco and Burden (1997). Tin and Jeffries (2001a) also found a ratio of 29.6 for ice keel area/ice sail area, and a ratio of 10.4 for ice keel area/snow sail area. For our corresponding 3D dataset, our ice keel volume/ice sail volume
ratios range from 11.6-19.0 and our ice keel volume/snow sail volume ratios from 4.6-6.4. Our ratios are somewhat lower than those of Tin and Jeffries (2001a), perhaps because drill line measurements of snow/ice freeboard tend to be biased low due to selection bias.

It may also be possible to infer the level ice thickness given a measured sail height. Tucker III et al. (1984) found the thickness of the level ice forming a sail ($L$) and its sail height ($S$), assuming buckling failure, could be related as $S \propto L^{0.5}$. Tin
and Jeffries (2003), following Melling and Riedel (1996), assumed that the sail height ($S$) could be related to the keel depth ($H$) as $H = 5S$, and thus the keel depth could be related to the level ice thickness as $H = aL^{0.5}$, and found $a = 5$ for a dataset from the Ross Sea. This coefficient of 5 is lower than the coefficients (15-20) for a variety of Arctic ridges in the Beaufort Sea (Tucker III et al., 1984; Melling and Riedel, 1996). When fitted to our PIPERS AUV dataset, we get $a = 6.7 \pm 0.7$. Note that here we use the mean draft of the level ice from the AUV dataset, which is very close to the mean thickness (and indeed, for
early winter, over level ice, the $F = D$ assumption should be approximately true). Following Leppäranta and Hakala (1992), Tucker III et al. (1984) and Timco and Sayed (1986), which found the range for the exponent could not be narrowed beyond beyond 0.5-1.0, we also try fitting a linear regression (with no intercept), giving $a = 9.3 \pm 1.5$. We expand this regression to include our full AUV dataset (20 scans, see Section 2) spanning a much wider range of keel depths (Fig. 5). We obtain $a = 6.8 \pm 0.4$ for the square-root relationship and $a = 6.5 \pm 0.7$ for the linear relationship. As the scatter is high, we select the
best model by choosing the lowest AIC (Akaike Information Criterion, see Akaike (1974)) as the $R^2$ is not well-defined for a fit with no constant term. In both the PIPERS-only and full-AUV datasets, the square-root relationship was a better model than the linear relationship, even if an intercept was included in the linear regression. Our coefficient of $a = 6.8$ is similar to the coefficient of 5 from Tin and Jeffries (2003), and both of these are much lower than the coefficients found for Arctic ridges (15-20), suggesting a possible morphological difference between Arctic and Antarctic ridges. We also performed a monomial





fit to identify the best exponent of $L$, which gave $H = 6.4L^{0.38}$. This had a marginally smaller AIC than the square-root fit, although this exponent is not within the range of 0.5-1 suggested by Timco and Sayed (1986) and Tucker III et al. (1984). In any case, both the square-root and monomial fits have considerably lower AICs than the linear fit, which suggests that the exponent is likely closer to 0.5 than 1.0.

This relationship could potentially be used the other way, by measuring the extreme value (sail height) and inferring the level ice thickness. Sail heights and deformed ice proportion are recorded when taking underway observations (ASPeCt) to estimate ice thickness, and so our empirical relationship may be applied to these data to give a basic estimate of sea ice volume in the Ross Sea. Worby et al. (2008) identified a relationship for estimating sea ice thickness, assuming a triangular sail and keel, as $T = 2.7RS + Z_u$ for some deformed proportion $R$, sail height $S$ and level thickness $Z_u$. This relationship was derived from

drilling data by initially working out the snow sail mean to draft mean ratio as 5.1, and then correcting for snow obscuring the deformed surface area to obtain a corrected ratio of 4.4. Using our own values for mean ice draft/freeboard, we coincidentally get the same corrected ratio of 4.4, and so we could expect the relationship above to hold for our dataset. However, our ridges are non-triangular, so we are not surprised to find that this relationship does not really hold with our PIPERS dataset, with a MRE of 56% (using the 99th percentile values for the sail height, which would be closer than the 100th percentile value to the

effective height of an equivalent-volume triangular sail). Further analysis using surface roughness as a variable to estimate ice thickness is detailed in Sections 3.1.2 and 3.2.3.

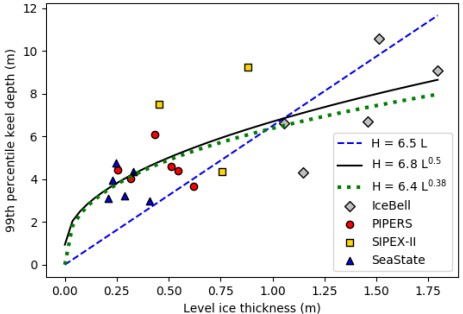

**Figure 5.** Level ice (draft) thickness vs keel depth (defined as the 99th percentile draft), following theoretical relationships described in Tucker III et al. (1984); Tin and Jeffries (2003). The square-root fit (black) has a much lower AIC (75.9) than the linear fit (blue, AIC = 92.7), and the monomial fit (green) has a slightly lower AIC (75.4) than the square-root fit. The mean relative errors (MRE) in predicting keel depths compares similarly, with MREs of 41%, 24% and 22% for the linear, square-root and monomial fits.

### 3.1.2   Estimating mean thickness with surface roughness

It is reasonable to expect that rougher ice, which is generally older and more deformed, should be thicker. Tin and Jeffries (2001b) found a linear relationship between the large-scale (1 m resolution, over 150 m) RMS roughness (i.e. the standard

deviation, $\sigma$) of the snow surface and its thickness, and also a linear relationship between the $\sigma$ of the surface and $\sigma$ of the draft, for 3 Ross Sea datasets from summer and autumn. Taking their ratio, we can estimate the linear relationship between the





$\sigma$ of the draft and the ice thickness, which we can compare to our AUV data (Fig. 6). Tin and Jeffries found that the survey-wide mean thickness was 5.5 times the survey-wide snow-surface roughness, and their snow-surface roughness was 1/3.7 of the ice bottom roughness. So, approximately, the survey-wide mean thickness would be 5.5/3.7 times the ice bottom roughness, giving a factor of 1.5. As not all our AUV datasets have corresponding lidar/snow data, we use the mean draft as an estimate of the

mean thickness. We find the same ratio of (survey-wide mean thickness)/(survey-wide bottom roughness) = $1.5\pm0.1$ for our AUV dataset (Fig. 6). Here, our survey-wide scale is very close to the floe scale of Tin and Jeffries (2001b), but our surveys are not necessarily representative of the whole floe due to deliberate selection of deformed ice. To avoid confusion, we refer to our large-scale statistics as survey-wide instead of floe-wide; each survey (e.g. PIPERS) is comprised of multiple scans (e.g. PIP4, PIP7), from which smaller windows are taken.

Due to the seasonally- and regionally-varying snow cover, the relationship between snow-surface roughness and bottom roughness is unlikely to be consistent across different climatologies, so we only analyze the thickness/(snow-surface roughness) ratio for the four full ice stations from Fig. 4 in the PIPERS dataset. This gives a ratio of $8.2\pm0.5$, with a mean relative error of the mean thickness of 12%, which is higher than the ratio of 5.5 from Tin and Jeffries (2001b). In this paper, we use the mean relative (percentage) error (MRE) of the predictions instead of the mean absolute error to prevent weighting prediction

errors in thicker ice differently to thinner ice, which is important as the majority of sea ice surfaces are thinner, undeformed ice. The ratio of (survey-wide surface roughness)/(survey-wide bottom roughness) is $1/(6.5\pm0.5)$, which is also different to the corresponding ratio of 1/3.7 from Tin and Jeffries (2001b). Interestingly, combining them to get a ratio of (survey-wide mean thickness)/(survey-wide bottom roughness) gives a very similar value of $1.3\pm0.1$, which agrees well with their value of 1.5. This may be because our scans captures larger features than a drill line can, and these may have different roughness values.

Moreover, drill lines may suffer sampling biases as previously discussed.

We repeat the analysis using local snow surface roughness ($\sigma$ of a 20m x 20m window at resolution 0.2 m) and local mean draft (Fig. 6b) instead of scan-wide statistics. The window size was chosen by using the range of the semivariogram for the floes (25 m), which we expect to represent the maximum feature length scale. This compares well to an average snow feature size of 23.3 m from early-winter Ross Sea drill lines from Sturm et al. (1998). We chose 20 m windows to balance this with the need

for a smaller window size to ensure a larger number of windows (= data points) for our analysis. The MREs using the snow surface roughness to predict mean local thickness range from 23-37% when fitting for each survey separately. For comparison, we also try fitting the local mean thickness to the local draft roughness, which has higher MREs of 31-48%. In general, rougher surfaces correspond to thicker ice, although the nature of this relationship may be nonlinear at higher resolutions.

Similarly, we may expect rougher areas to trap more snow (Massom et al., 2001; Kwok and Maksym, 2014). Although

Kwok and Maksym (2014) averaged the snow depth and surface roughness over a much larger scale (4 km scale at resolutions 1-10 m), we also find snow accumulates preferentially in areas of deformation. We find that snow depth at a 20 m scale can be approximated by a linear function of the surface roughness (slope: 0.80, intercept: 0.12 m). This is a similar relationship to what they found (slope: 0.83-1.25, intercept: 0.07-0.18 m), despite their dataset being from a different region (Weddell/Bellinghausen Seas) and season (spring) than ours (Ross Sea/winter). Our correlation ($R = 0.66$) is also comparable to theirs ($R = 0.71$). It

is not surprising that snow accumulates in areas of deformation, but the relatively high scatter in using a simple linear model





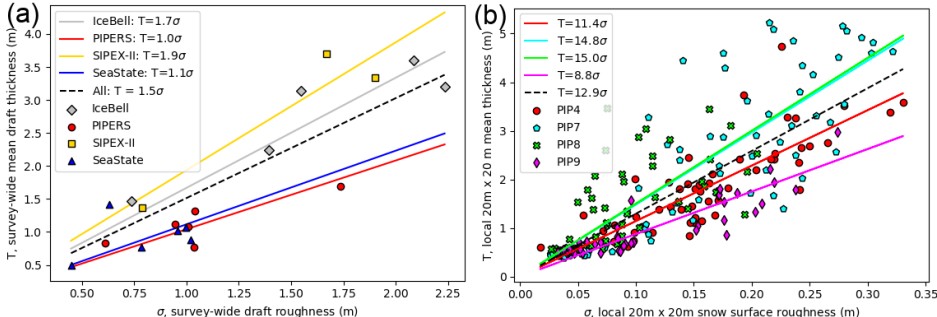

**Figure 6.** (a) Survey-wide RMS roughness ($\sigma$) vs. floe mean draft thickness for the different AUV datasets, to be compared against a slope of 1.5 from Tin and Jeffries (2001b). Our fit for all data (black dotted line) also has a slope of 1.5. The resolution is 0.5 m. PIPERS and SeaState largely focused on first-year ridges, whereas Icebell data is from consolidated late spring (potentially with multi-year ice), and SIPEX-II is from early spring. Fits to the individual datasets are color-coded. The MRE in the predicted mean thicknesses are 11%, 17%, 12%, and 18% for IceBell, PIPERS, SIPEX-II and SeaState respectively, and 33% for all data. (b) Local $\sigma$ of the snow surface vs. local mean thickness, for PIPERS data only. The MREs for predicting mean local thickness range from 23-37%, with the fit for all PIPERS data (black dotted line) having a MRE of 33%. This is slightly lower than the MRE of 49% for predicting mean draft thickness using local draft roughness from PIPERS (not shown).

motivates more advanced techniques to analyze deformed surfaces. RMS roughness does not account for spatial features, as any permutation of points within a grid would have the same standard deviation.

These results relating ridge morphology to other metrics largely agree (with some exceptions) with prior literature, despite the difference in resolutions and ridge thicknesses. This suggests that surface morphology can be related, at least to a limited extent, to sea ice thickness, and potentially improve on simple linear predictions of ice thickness. This is discussed in further in Section 3.2.3.

## 3.2 Estimating thickness using hydrostatic balance

To accurately calculate sea ice thickness without making assumptions of snow distribution, we need to use combined measurements of ice draft (AUV), surface elevation (lidar) and snow depth (probe). Here, we primarily use PIPERS data to focus on early-winter floes, and also because this is the largest such dataset from the same season/region, which is important so that the ridges have consistent morphology. The lidar and AUV data were corrected by aligning the mean measurements of the level areas of the drill line. It is important to use the level areas only as drill line measurements are likely to be biased low due to the difficulties of getting the drill on top of sails, and the presence of seawater-filled gaps that may be confused with the ice-ocean interface.

Assuming hydrostatic balance, Eq. 1 should hold for all datasets. However, drill lines have coarse resolution, which may not capture the local variability in elevation between drill points, and so drill points may not even be in hydrostatic balance. Drill lines are also only 2D, and so surface variations in the 3rd axis may not be accounted for in drilled points. Moreover, densities



may vary spatially, in particular around deformed surfaces that may contain air/water gaps. Due to the difficulty in drilling sail peaks, freeboards from drill lines may also be undermeasured. Our 3D data should therefore more accurately sample ridged regions, and we should expect hydrostatic balance to hold.

Ozsoy-Cicek et al. (2013) compiled various Antarctic datasets to investigate the relationship between sea ice thickness, snow
depth and surface elevation. Assuming hydrostatic equilibrium is reached over the window size (20 m), we use Eq. 1 to fit a regression for sea ice thickness ($T$) as a linear function of surface elevation ($F$), sometimes along with snow depth ($D$). This approach has been applied by Zwally et al. (2008); Worby et al. (2011); Yi et al. (2011); Xie et al. (2013) over a variety of scales.

### 3.2.1 Fitting to surface elevation only

Although we have snow depth measurements in addition to surface elevation measurements, in general there are far fewer snow data and so we first try to fit with just surface elevation, by making some snow depth assumptions.

For level topography, where the snow = surface elevation assumption is supposedly valid (set $F = D$ in Equation 1), Eq. 1 would simplify to $T = 2.7F$ (using density values from Zwally et al. (2008)). In contrast, when the topography is sufficiently rough, there is considerable ice freeboard, which may even exceed snow depth. If we assume the snow is negligble ($D = 0$),
which may be the case at the sail peak, Eq. 1 becomes $T = 9.4F$. These values become lower and upper bounds for fitting $k$ in $T = kF$.

All our coefficients, which range from 2.9-6.1, fall between these two extremes of snow-only freeboard and ice-only freeboard, with the lowest value of 2.9 being for level topography, as expected. Our sampled areas are likely not representative of typical area-averaged deformation rates of sea ice due to these survey areas being selected for their heavy deformation, and so
the fitted coefficients for individual floes and the "All" category are considerably higher than 2.7. In contrast, the coefficients for $F$ of 2.8-3.0 in Xie et al. (2011), and 2.2-3.1 in Ozsoy-Cicek et al. (2013), suggest that at floe-wide and larger scales, there is enough level ice that the snow = surface elevation assumption is valid, at least for this region/season. It is also possible that these drill lines have undersampled ridged ice. Our coefficient of 5.79 is much higher, which suggests that there is some non-zero component of ice freeboard in the surface elevation measurements. For example, if we assume typical snow/ice densities
of 300 kgm$^{-3}$/920 kgm$^{-3}$, we can estimate that snow, on average, comprises 54% of the measured surface elevation, which means Eq. 1 simplifies to $T = 5.8F$, as in Fig. 7. In further support of this, our dataset has mean snow depths for the four surveys ranging from 16-26 cm, and mean surface elevations ranging from 24-37 cm, implying non-zero mean ice freeboards from 6.5-11 cm. If the proportion of ice to snow were constant (and their effective densities, too), then the best-fit line would have no scatter. This is not the case in Fig. 7, and indeed the standard deviation of ice freeboard across all windows was 7.9 cm
(mean: 9.0 cm). This means that assuming a constant snow/ice density or a constant snow/ice proportion is not justified, and hence it is likely that simple statistical models break down when looking at deformation on a small scale, or when large-scale snow deposition and ice development conditions vary.

Note that our sampled region is not representative of the floe, as we have deliberately chosen a heavily-deformed area, and so the amount of ice freeboard is higher in our survey, in contrast to large-scale averaged surface elevations like Xie et al. (2013)

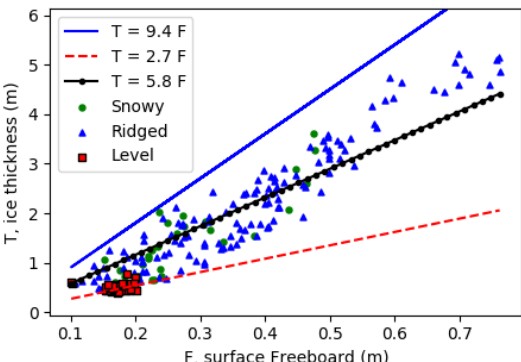

**Figure 7.** The sea ice thickness ($T$) as a function of measured surface elevation ($F$). As expected, all points lie between the two extreme regimes (no ice freeboard and no snow freeboard). The level surfaces mostly have no ice freeboard, as expected, though there is some scatter that suggests a varying component of ice freeboard. The best fit line for all windows from Table 2 is shown in black. Assuming mean snow and ice densities of 300 and 925 $\mathrm{kgm^{-3}}$, this implies a mean proportion of 62% snow and 38% ice in the surface elevation. Again, the scatter around the best fit line indicates that this proportion is changing. Some points for the level category fall below the $T = 2.7F$ line, suggesting that snow densities in these areas are $<300\,\mathrm{kgm^{-3}}$ (or effective ice density $<915\,\mathrm{kgm^{-3}}$.)

and Ozsoy-Cicek et al. (2013). This suggests that at large scales for some seasons/regions, it may be reasonable to assume that the mean freeboard is zero, but this is not the case at smaller scales.

Ozsoy-Cicek et al. (2013) found the fitted linear regression between $T$ and $F$ as $T = 2.45F + 0.21$ for a winter Ross Sea dataset. With our PIPERS dataset, our equivalent fit is $T = 7.67F - 0.73$, with 23% MRE. Using the relationship from Ozsoy-
Cicek et al. (2013) on our dataset, the MRE is 36%, and the error in estimating the overall survey mean thickness is 41%, despite being from the same climatology. This means that relationships from other datasets from the same region/season do not generalize well, especially if the proportion of deformed ice (and hence nonzero freeboard) is significant.

It is possible to interpret our negative intercept as a bias due to fitting a linear model across two roughness regimes. From above, the two regime extremes (no-ice vs. no-snow contribution to surface elevation) give coefficients of 2.7 and 9.4 for $F$.
In general, we expect the proportion of ice freeboard to gradually increase as $F$ increases from thinner, level ice to thicker, deformed ice. Although snow also accumulates around deformed ice, there may also be local windows at parts of the ridge with no snow (e.g. the sail). Fitting one line through these two clusters of points would result in a coefficient for $F$ between 2.7 and 9.4 and a negative intercept, which we find in almost all our cases. The one exception is the no-intercept fit with $F$ for the level category, which is essentially a null fit (as over 90% of the thickness values are clustered around 0.5 m, with surface elevations
varying over a narrow 5 cm range). In contrast, the coefficients for $F$ from Ozsoy-Cicek et al. (2013); Xie et al. (2011) are all $\sim$3, because these studies average over multiple floes and have a sufficiently small proportion of deformed surface area to assume a negligible mean freeboard. This could also explain why their intercepts are positive.



### 3.2.2 Fitting to surface elevation and snow depth

Using typical values of 910 kgm$^{-3}$ for ice density, 1027 kgm$^{-3}$ for water density and 323 kgm$^{-3}$ for snow density from Worby et al. (2011), the coefficients for the freeboard $F$ and snow depth $D$ should be 8.8 and 6.0. Similarly, Zwally et al. (2008) used corresponding densities of 915.1 kgm$^{-3}$, 1023.9 kgm$^{-3}$ and 300 kgm$^{-3}$, giving a freeboard coefficient of 9.4

and a snow coefficient of 6.7. We compare these to our results of the (multi)linear regressions in Table 2. We include fits with an additive constant, even though this is unphysical, to see how well a linear model can predict SIT. Our coefficients when fitting over all 4 floes are 10.4 for $F$ and 6.8 for $D$, which are comparable to those inferred from Zwally et al. (2008), although there is considerable variation between the floes (7.9-10.6 for $F$; 3.9-6.3 for $D$). As discussed below, this suggests a lack of generalization in the fits. Assuming the density of seawater is fixed at 1028 kgm$^{-3}$, this gives bounds for the effective densities

and standard errors of sea ice and snow as 929.4±3.5 kgm$^{-3}$ and 356.3±57.2 kgm$^{-3}$. The snow density is in line with Sturm et al. (1998), which found mean densities of 360-390 kgm$^{-3}$ during winter in the Ross Sea, as well as the measured snow densities from PIPERS (245-300 kgm$^{-3}$). The ice (effective) density errors here are only for our PIPERS dataset and may not apply to other samples from the Ross Sea in winter as the effective density is affected by the proportion of ridged ice. Moreover, it is important to note that under this fitting method, the density estimates are coupled (due to $\rho_i$ appearing in both coefficients

in Eq. 1) and if the estimate of $\rho_s$ decreases, $\rho_i$ increases. For example, if $\rho_i$ = 935 kgm$^{-3}$ (unusually, but not impossibly high for the effective density of ridged ice, which includes some proportion of seawater - see Timco and Frederking (1996), and also note that this is the effective density, including some proportion of seawater), the best estimate for $\rho_s$ becomes 312 kgm$^{-3}$, which is closer to

Although each individual floe has MREs of 10-20% when fitting with an intercept, the large variations in coefficients and
constants suggests that the linear model does not generalize well between floes. For example, using the relationship from PIP7, PIP8 or PIP9 on PIP4 gives 30%, 33% or 35% MRE respectively, compared to 10% error for the PIP4 coefficients; using the PIP4, PIP7, or PIP8 fits on PIP9 gives 67%, 70% or 43% mean error, compared with 21% for using PIP9 coefficients. Table 3 summarizes the fit and test errors for using each of the floes as the test set. The fit MREs range from 17-20% (fitting with constant) and 25-36% (without), and the test MREs range from 23-34% (with constant) and 12-59% (without). Using typical
values for snow/ice density from literature mentioned above (giving $F$ and $D$ coefficients of 8.8 and 6.0 (Worby et al., 2001) or 9.4 and 6.7 (Zwally et al., 2008)) gives MREs of 26% in both cases, and errors in estimating overall mean thickness of ∼ 15%. The high variability in the test errors suggests that statistical relationships may not generalize to future datasets, even those from the same climatology. This is an important limitation of applying empirical fits from small datasets.

### 3.2.3 Incorporating surface roughness into the fit

Given that we expect effective density variations for different surface types, we expect SIT estimates to improve with the addition of surface morphology information. The most simple of these is the surface standard deviation, as prior studies have found that this is correlated to the snow depth (Kwok and Maksym, 2014), and we have previously shown that the surface $\sigma$ has some prediction power for the mean thickness (Fig. 6b). Adding the roughness as a variable to the fit, so that





**Table 2.** Fitted coefficients for sea ice thickness $T$ as a multilinear regression of the surface elevation $F$ and snow depth $D$ (Section 3.2.2), and also fitting for $F$ only (Section 3.2.1). Surfaces are also categorized (Fig. 8) to incorporate roughness into the fits (Section 3.2.3). As the $R^2$ is not well-defined for a fit with no constant term, the Akaike Information Criterion is used for model selection (Akaike, 1974). The $R^2$ is reported for the with-constant fits only and is adjusted for the different sample sizes in each fit. For each dataset, the smallest AIC value is **bolded**, and the second-lowest underlined. For individual floe fits, only PIP8 is shown for brevity as the other floes have comparable errors/coefficients.

|  |  | $R^2_{adj}$ | AIC | MRE, m [%] | F coeff. | D coeff. | Constant |
|---|---|---|---|---|---|---|---|
| PIP8 | Both, no int. | N/A | 37.3 | 0.26 [24] | 9.03 ± 1.0 | -5.45 ± 1.25 | N/A |
|  | F only, no int. | N/A | 52.1 | 0.32 [31] | 4.81 ± 0.15 | N/A | N/A |
|  | Both | 0.92 | **5.30** | 0.18 [15] | 8.85 ± 0.73 | -2.70 ± 1.02 | -0.70 ± 0.11 |
|  | F only | 0.91 | 10.2 | 0.20 [16] | 7.07 ± 0.30 | N/A | -0.81 ± 0.10 |
| Ridged | Both, no int. | N/A | 111 | 0.29 [22] | 10.33 ± 0.44 | -6.53 ± 0.67 | N/A |
|  | F only, no int. | N/A | 182 | 0.38 [30] | 6.09 ± 0.10 | N/A | N/A |
|  | Both | 0.94 | **75.5** | 0.25 [17] | 10.42 ± 0.39 | -5.06 ± 0.63 | -0.45 ± 0.07 |
|  | F only | 0.91 | 128 | 0.31 [21] | 7.59 ± 0.20 | N/A | -0.65 ± 0.08 |
| Level | Both, no int. | N/A | -56.5 | 0.07 [13] | 3.58 ± 0.77 | -0.82 ± 0.96 | N/A |
|  | F only, no int. | N/A | -57.7 | 0.07 [14] | 2.92 ± 0.10 | N/A | N/A |
|  | Both | 0.07 | **-72.3** | 0.06 [12] | 0.87 ± 0.85 | -1.22 ± 0.76 | 0.52 ± 0.11 |
|  | F only | 0.00 | -71.6 | 0.07 [13] | 0.02 ± 0.67 | N/A | 0.50 ± 0.11 |
| Snowy | Both, no int. | N/A | 36.4 | 0.29 [34] | 10.45 ± 1.37 | -6.29 ± 1.63 | N/A |
|  | F only, no int. | N/A | 47.7 | 0.36 [41] | 5.22 ± 0.25 | N/A | N/A |
|  | Both | 0.87 | **19.9** | 0.22 [23] | 11.88 ± 1.15 | -5.33 ± 1.33 | -0.63 ± 0.14 |
|  | F only | 0.81 | 32.3 | 0.27 [24] | 7.74 ± 0.59 | N/A | -0.72 ± 0.16 |
| All | Both, no int. | N/A | 194 | 0.30 [31] | 10.42 ± 0.37 | -6.81 ± 0.53 | N/A |
|  | F only, no int. | N/A | 318 | 0.41 [44] | 5.79 ± 0.09 | N/A | N/A |
|  | Both | 0.94 | **109** | 0.24 [20] | 10.19 ± 0.31 | -4.51 ± 0.49 | -0.52 ± 0.05 |
|  | F only | 0.92 | 179 | 0.28 [23] | 7.67 ± 0.15 | N/A | -0.73 ± 0.05 |



**Table 3.** A compilation of the MRE of different fitting methods. Coefficients for the linear fits are shown in Table 2 and details are in Sections 3.2.1-2. The leftmost column indicates the floe that was excluded from the fitting data (e.g. the first row indicates fits that were done over the PIP7-9 data and then tested on PIP4). The validation error was used for comparison with the linear model fits, as the training error can be made artificially low by overfitting. On average, the ConvNet (Section 3.3) achieves the best generalization in the fit, even though there are individual anomalous cases. For example, the linear, no-constant fit using PIP4 as a test set has a low test error of 12% despite having a high fit error of 36%. This is simply coincidental in that the scatter for the fit is so high that the best-fit coefficients end up close to the corresponding coefficients for the PIP4 floe. The F-only (no constant) column is included as this is directly comparable to our ConvNet method, as they use surface elevation as the only input (no snow depth) and maintains the zero freeboard = zero thickness condition.

| Test set | Linear (no constant) | | Linear (with constant) | | F only (no constant) | | ConvNet | |
| --- | --- | --- | --- | --- | --- | --- | --- | --- |
| | Fit MRE | Test MRE | Fit MRE | Test MRE | Fit MRE | Test MRE | Val. MRE | Test MRE |
| PIP4 | 36% | 12% | 17% | 31% | 51% | 13% | 14% | 20% |
| PIP7 | 25% | 33% | 20% | 24% | 35% | 45% | 14% | 18% |
| PIP8 | 33% | 32% | 22% | 23% | 45% | 57% | 16% | 20% |
| PIP9 | 27% | 59% | 20% | 34% | 40% | 74% | 14% | 20% |
| **Average** | **30%** | **34%** | **20%** | **28%** | **43%** | **47%** | **15%** | **20%** |

$T = c_1 F + c_2 D + c_3 \sigma (+c_0)$, gives slightly lower fit MREs of 16-20% (with constant) and 24-34% (without) and slightly higher test MREs of 14-28% (with constant) and 28-58% (without). This is not much of an improvement, and it is possible that $\sigma$ is too simplistic a metric to improve the fit. Furthermore, there is no particular reason to expect the surface $\sigma$ to be linearly combined with the snow depth and surface elevation, even if it makes dimensional sense.

Instead, we can try using the roughness as a regime selector. To do this, firstly the lidar windows were classified manually into snowy surface, level surface, ridged surface and deformed surface categories (Fig. 8). If it had both a ridge and snow, it was classified as ridged. 'Deformed' was intended as a transitional category for images that had no clear ridge but were generally rough - this comprised, typically, $\sim 5\%$ of an image and was excluded from analysis. The snowy, level and ridged categories were individually fitted to see if there were any differences in the coefficients; these are also reported in Table 2.

We then used a two-regime model over all four floes, so that ice thicknesses for the low-roughness surfaces are estimated using the 'level' coefficients, and high-roughness surfaces using the 'ridged' coefficients. This resulted in MREs of 16-21% (with constant) and 17-19% (without), assuming 20-50% of the surface is deformed. This is slightly better than for fitting the 'all' category in Table 2 (20% MRE with constant and 31% without), suggesting that distinguishing topographic regimes improves thickness estimates. However, this fit has issues with generalizing to other floes. If the fit for the rough/level coefficients

is done using only 3 floes and then applied to the remaining floe (using a surface roughness threshold determined from that floe, and again assuming 20-50% of the surface is deformed), the test MREs averaged over all possible choices of test floe are considerably higher (19-25% when fitting with constant, 34-36% when fitting without, and in each case averaging the results over all possible test floe choices). This does not improve on the generalization if not using a two-regime model, where the test

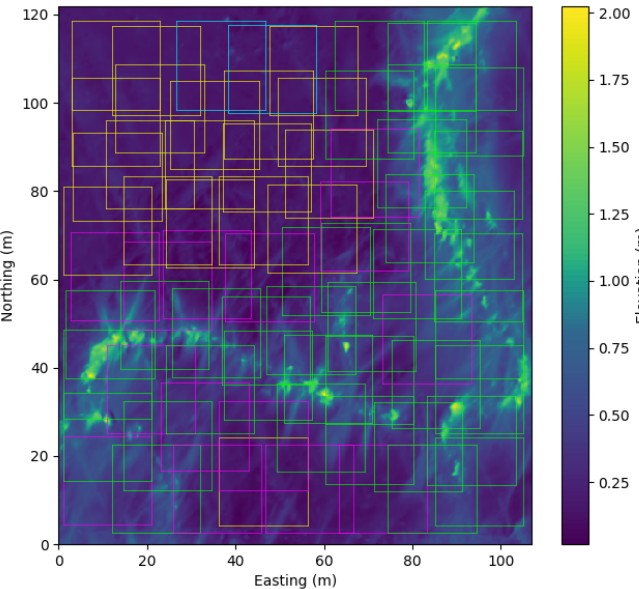

**Figure 8.** An example lidar scan from a station (PIP4) with the classified segments. Yellow = level surface, green = ridged surface (possibly with snow), magenta = snowy surface and blue = deformed surface (excluded from analysis). Snow features are clearly visible emanating from the L-shaped deformation.

MREs, again averaged over all test cases, are 28% (with constant) and 34% (without). Although the distinguishing of regimes may improve the model fits, it does not improve the test errors, again because this is likely too simplistic.

In reality, there is no reason why ice thickness should be non-zero given a zero surface elevation and snow depth. For all cases, the AIC and mean error is lower when fitting with an intercept, but a negative intercept, as in all our fits, sets a lower
bound on values for $F$ and $D$ that return non-negative values of $T$. This reduces the applicability of these statistical models for thin ice. Taking the zero-intercept models only, estimating the survey-wide thickness has relatively low error (10-20%), but estimating local thickness has much higher errors (30-50%), which means that variations in ice thickness, such as those around deformation areas, cannot be precisely estimated. This motivates more advanced techniques to decrease the MRE, especially those that also maintain the physicality of zero surface elevation = zero thickness. Given that high surface elevation values may
be either snow features or ridges (leading to very different ice thicknesses), we need to distinguish these surface features in a non-arbitrary way. In particular, we want to account for the complex deformation morphology, which we expect to be better predictors of thickness than the simplistic metrics used previously.

### 3.3 Predicting SIT with deep learning

One advantage of deep learning techniques is that they are able to learn complex relationships between the input variables and a
desired output, even if the relationships are not obvious to a human. Although they are commonly used for image classification purposes, they can also be used for regression (e.g. Li and Chan, 2014). We expect a convolutional neural network (ConvNet)





to achieve lower errors in estimating sea ice thickness, as they are able to learn complex structural metrics, in addition to simplistic roughness metrics like $\sigma$. Our input is a windowed lidar scan (surface elevation) and an output of mean ice thickness. Notably, there is no input of snow depth, nor any input of ice/snow densities. This allows the ConvNet to infer these parameters by itself, and more importantly, to potentially use different density values for different areas.

Our best-performing linear regression has a mean error of 23%, though this includes an unphysical intercept, and also does not generalize to other datasets from the Ross Sea. We seek our model to improve on this error rate, while also being generalized enough to apply to different floes (from the same climatology), and also maintaining the physicality of zero surface elevation = zero thickness. Due to our limited dataset, comprised entirely of Ross Sea data in early winter, we do not expect our results to necessarily apply to other regions or seasons, which may have different snow distributions, ridging frequencies, or other causes

of morphological differences. We intend simply to demonstrate the potential for these techniques to improve estimates of SIT. We do, however, expect our methods to generalize to new floes from the same region/season as our data.

    Full details for our ConvNet are given in the Appendix. The training set consisted of randomly-selected 20 m x 20 m windows from three PIPERS ice stations. We chose 20 m windows, as in Section 3.1, by inspecting the semivariogram. 20% of the data were excluded from training so that it could be used as a validation set. The remaining floe was kept as a test set,

in case the training and validation windows had similar morphology and the validation set was thus not entirely independent of the training set. To prevent cherry-picking, the CNN was trained four times, with a different floe used as the test floe each time. These results are shown in Table 3. Although the training error is directly analogous to the fit error for linear models for some dataset, it is much easier to overfit with a ConvNet as the training error can be made arbitrarily low. As a result, we compare our validation error to the linear fit errors, and also use our test errors as a test of the generalization of our model.

From here onwards, analysis of the ConvNet refers to the one using PIP8 as a test set, though using a different one would yield qualitatively similar analysis.

    The input windows were randomly flipped and rotated in integer multiples of $90^o$ to help improve model generalization, and dropout layers ($p = 0.4$) were added after the first and second convolutional layers to reduce overfitting (Srivastava et al., 2014). The best validation error of 15.5% occurred at epoch 881, corresponding to a training error of 11.3% (Fig. 9). The mean

test error (on the excluded floe) was 20.0%. Although the linear models have a similar fit error, they do not generalize as well to the test set, and the resulting thickness distribution is visibly different to the real test distribution.

    This shows better generalization than the linear models (test MREs from 28-47%). Although the best-performing linear models have only slightly higher test MREs (23-24%) than our ConvNET (20%), the range of errors is much greater, with test MREs of 23-34%, whereas the ConvNet has remarkably consistent test MREs of 18-20%. Furthermore, it is important to

remember that achieving these comparably low MREs with linear models requires snow depth as a variable and also a constant term. These constants, as shown in Table 2, are all negative, which means $T < 0$ for $F = D = 0$ which is clearly unphysical. The fits to surface elevation only, which is using the same input data as the ConvNet, have considerably higher MREs (13-74%). For sake of comparison to models that use RMS error such as Ozsoy-Cicek et al. (2013), the RMS errors for our validation and test datasets were 6-11 cm, which is considerably lower than the RMS errors of $> 50$ cm for single drill point measurements

in drill lines given in Ozsoy-Cicek et al. (2013), and also lower than the RMS error of 26 cm from using one varying-density


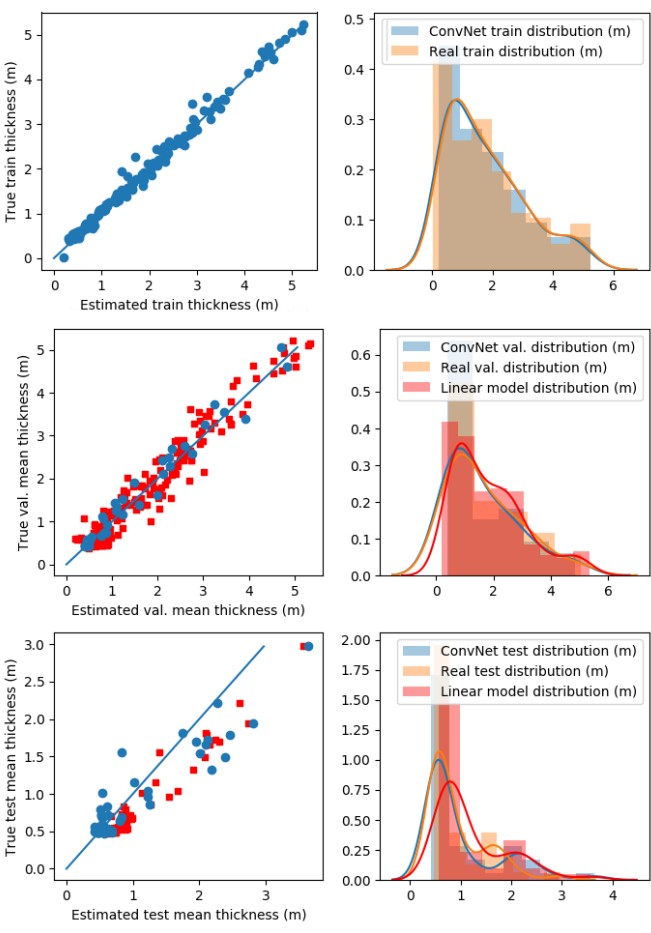

**Figure 9.** ConvNet results. The top panels show the learned model applied to the training data (80% of randomly sampled 20m x 20m windows from PIP4, PIP7, PIP8), with MRE 12%; the middle panels show the learned model applied to the validation data (remaining 20% of the randomly sampled 20m x 20m windows from PIP7-9) with MRE 16%; the bottom panels show the learned model applied to randomly sampled 20m x 20m windows from PIP9, as a check against learning self-similarity, with MRE 19%. The panels on the right show the resulting thickness distribution, both as a histogram and as a continuous function. We also show the best linear model fitted for the PIP4, PIP7, PIP8 in the middle panels (red) which has a comparable MRE of 20%, and the results of this model applied to PIP9 (test floe) in the bottom panels (red) (MRE: 34%). Our results suggest slight overfitting, as the test scatter is higher than the training scatter, but the learned model still generalizes fairly well, with MREs much lower than linear models, even when including an unphysical intercept to improve the fit (Table 3).

layer at a 70 m scale from Li et al. (2018). Note that the RMS error is not linked to the surface RMS roughness, which is just the standard deviation of the surface elevation.

It is not entirely fair to compare the single-point RMS errors to our survey-averaged RMS errors due to the different length scales; a better comparison is in estimating mean scan-wide thicknesses (∼100 m length scale). This error should not be





confused with the mean error in estimating the local window means, which is what is being optimized by the ConvNet. The mean error in estimating survey-wide thickness (averaging through all 4 datasets) is 5% for the training set, 6% for the validation set and 11% for the test set. The 5-11% MREs for the scan-wide mean thickness predictions correspond to RMS errors of 1-3 cm, which is much lower than the best-performing linear regression for a Ross Sea floe-wide dataset from Ozsoy-

Cicek et al. (2013) with an RMS error of 16 cm.

As shown in Fig. 9, the ConvNet does seem to be capturing the thickness distribution of the test floe, even if the individual window mean estimates have some scatter. In contrast, the linear models have considerably different thickness distributions (Fig. 9) despite having only a slightly higher test MRE (Table 3). In any case, the key result of the ConvNet is in the significantly reduced error in the local (20 m scale) mean thickness (MRE of 15-20%), which also gives a low, $\sim 10\%$ error of

the average scan-wide thickness. Moreover, this high accuracy also carries over to new floes from the same climatology. An additional advantage of the ConvNet is that it does not require specifying snow/ice densities, but instead implicitly accounts for the (potentially spatially-varying) densities with its filters (discussed below). The ConvNet also gives an output thickness of $4 \times 10^{-2}$ m (essentially zero) when the input is a zero array, which is physically appropriate.

When applying this model to lidar inputs from a different expedition (SIPEX-II, see Maksym et al., in prep) with different

climatology (different season and different region), the MRE is 69%, and the error of predicting the survey mean is 51%. This suggests that other seasons/regions may have different relationships between the surface morphology and SIT, which is not surprising given that snow accumulates throughout winter. The SIPEX-II data was collected during spring in coastal East Antarctic in an area of very thick, late-season ice with very deep snow with large snow drift features of length scales >20 m (which would not be resolved by the ConvNet filters here). It is also possible that datasets from spring, such as SIPEX-II,

will not be as easy to train networks on because the signficantly higher amounts of snow may obscure the deformed surface. Although this points out a limitation of this method, which restricts any trained ConvNet to a narrow range of climatologies, it also adds weight to the idea that the ConvNet is learning relevant morphological features. A ConvNet trained on Arctic data would likely learn different features (e.g. melt ponds and hummocks), although additional filters may be needed to distinguish multi-year and first-year floes.

We also tried different inputs, such as using 10 m x 10 m windows, which had training/validation/test errors of 9%/18%/25%, and using 20 m x 20 m inputs with half the resolution (i.e. 0.4 m), which had errors of 7%/13%/25%. The smaller window case has a slightly higher validation error than the above ConvNet, and the coarser-resolution input has a slightly lower validation error than the above ConvNet, but both cases have slighly higher test errors. It is not surprising that a smaller window has a higher error, as the isostatic assumption may no longer be valid. Larger windows, which are more likely to capture surface

features, are likely to improve the fit, but our dataset is too small to test this as larger window sizes would mean fewer training inputs. However, it is promising that the validation errors are lower at a coarser resolution. This suggests that this method may indeed extend to coarser, larger datasets like those from airborne laser altimetry from OIB. We also tried training for the mean snow depth given the lidar inputs, with training/validation/test errors of 15%/17%/18%, which is very similar to the thickness prediction. This is not entirely surprising as, if hydrostatic balance is valid, being able to predict the mean thickness given some

surface elevation measurements naturally gives the mean snow depth via Eq. 1.





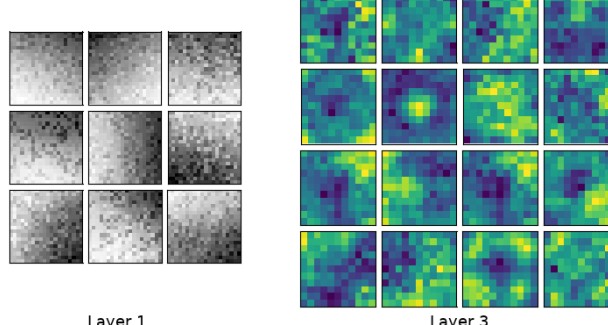

**Figure 10.** Typical weights learned in the first and last convolutional layers. Weights learned from the third layer are shown using the same colormap as the lidar in Fig. 8 to facilitate comparison. The filters in layer 1 correspond to edge detectors, and the filters in layer 3 may be higher-order morphological features like 'bumps' (snow dunes) and linear, strand-like features (ridges). The filter size of the first layer corresponds to 4.0 m and the third layer is 8.8m.

## 4    Discussion

### 4.1    ConvNet metric analysis

Although the ConvNet achieved a much lower test error than the linear fits, the inner workings of a ConvNet are not as clear to interpret. Here, we try to analzye the learned features by passing the full set of lidar windows through the ConvNet to see

if the final layer activations resemble any kind of metric. The below analysis of features is very qualitative, as it is inherently very difficult to characterize what a ConvNet is learning.

One helpful way to gain insight on what the ConvNet is learning is to inspect the filters. Filters in early layers tend to detect basic features like edges (analogous to a Gabor filter, for example), with later layers corresponding to more complex features like lines, shapes, or objects (Zeiler and Fergus, 2014). We see similar behavior in our filters; typical filters learned in our

model are shown in Fig. 10. Early filters highlight basic features like edges when convolved with the input array, while later filters show more complex features. These complex features are hard to interpret, but are clearly converged and not just random arrays. For example, a 'blob' feature could be a snow dune filter, while filters with a clear linear gradient could correspond to the edge of ridges. The filters in the final layer are around ∼8 m in size. This may be too small to resolve the entire width of the ridges in our dataset, but would be enough to identify areas near ridges. With a larger windowed lidar scan, such as those

from OIB with scan width ∼ 250 m (Yi et al., 2015), we expect better feature identification, as the entire width of a ridge can be resolved within a filter.

The learned weights for the final (8 x 1) hidden layer and their activations (when each input window is fed forward through the ConvNet) are shown in Fig. 12a, grouped by category (level, ridged, snowy). These should correspond to (unspecified) metrics, which are linearly combined with the weights shown in Fig. 12b. It is clear that level surfaces are distinguished from

ridged and snowy surfaces, but ridged and snowy surfaces show considerable overlap with each other. While it is not possible



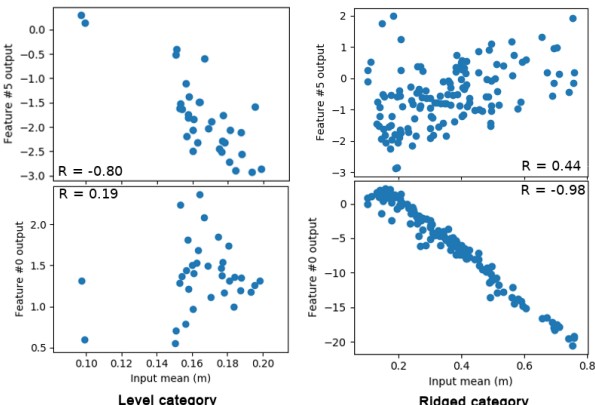

**Figure 11.** Scatter plot showing correlations between features and real-life metrics. Here, features #0 and #5 correlate strongly to the mean elevations of the level and ridged surfaces respectively, but not the other way around. This suggests that the level and ridged surfaces are treated differently, implying a different effective density of the surface freeboard. The correlation for the level category is not as strong; without the two points near $x = 0.1$, $|R| = 0.64$, so this feature is possibly a combination of the mean elevation and something else.

to determine with full certainty what each of the 8 features corresponds to, we can correlate these features to metrics that we may expect to be important for estimating the ice thickness and see which ones match. Doing this analysis, for ridged surfaces, features #0, #3 and #6 had a strong correlation ($|R| > 0.95$) to the mean elevation; for snowy surfaces, these three features had a slightly weaker correlation ($0.88 < |R| < 0.96$) to the mean elevation; and for level surfaces, features #1 and #5 had a

slight correlation ($|R| = 0.67$ and $0.80$ respectively) to the mean elevation (Fig. 11). However, features that correlated to the ridged surface mean elevation did not correlate to the level surface mean elevation, and vice versa. This suggests that the mean elevation for level surfaces is treated differently (e.g. given a different effective density) than other categories.

For ridged surfaces, in addition to the mean elevation, the RMS roughness was also important, with features #2 and #4 weakly correlating ($|R| = 0.61$) to the standard deviation of the window. The standard deviation had a slightly weaker correlation

($|R| = 0.55$) for level surfaces, and virtually none at all for snowy surfaces ($|R| < 0.20$). Another measure of roughness is the rugosity (the ratio of 'true' surface area over geometric surface area, see Brock et al. (2004)). This was most important for the snowy category, with $|R| = 0.57$ for feature #7, compared to $|R| = 0.53$ for feature #6 for ridged surfaces and $|R| = 0.22$ for feature #2 for level surfaces. As we found before, these features were much more strongly correlated to the mean elevation and standard deviation respectively for their respective surface category. This was not the case for feature #7 for snowy surfaces,

which had a similar correlation ($|R| = 0.54$) to the mean elevation and a much weaker correlation ($|R| = 0.35$) to the surface $\sigma$. To summarize, for all categories, the mean surface elevation is important (though weighted differently, as different filters are activating for different categories). For both level and ridged surfaces, the RMS roughness is important, and for snowy surfaces, the rugosity is also important. All the above analysis suggests that there are important regime differences for estimating sea ice thickness.





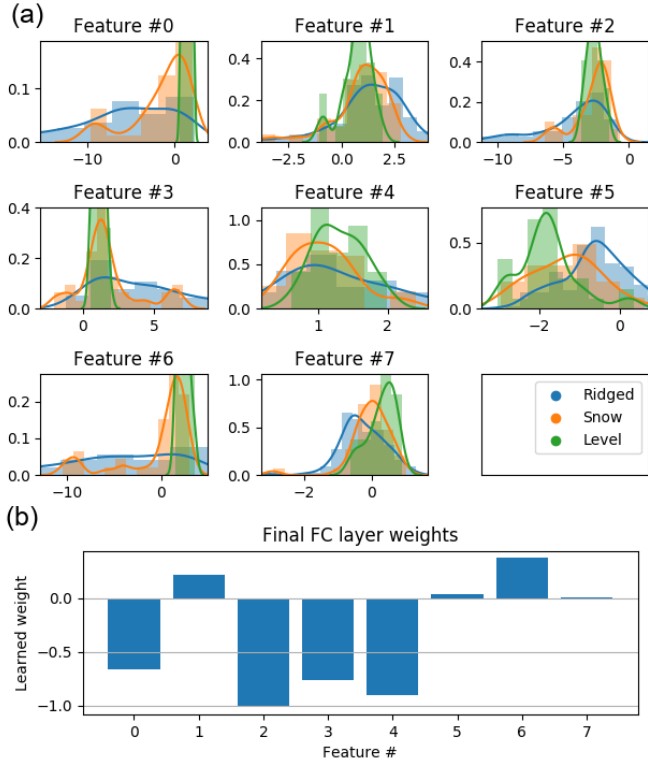

**Figure 12.** (a) Distribution of the final (8 x 1) layer activations for the level, ridged and snow categories from Fig. 8, and (b) the learned weights for the final fully-connected hidden layer. To generate the final thickness estimate, the activations in (a) are multiplied with the weights in (b), then summed.

This is by no means an exhaustive list, but it suggests that the ConvNet is learning useful differences between these categories. However, as suggested by the considerable overlap in the distributions in Fig. 9, these categories may also not be the most relevant classifications. Alternatively, a t-distributed Stochastic Neighbor Embedding (see Maaten and Hinton (2008)), which is an effective cluster visualization tool, shows that ridged and level surfaces are clearly distinguishable, but there is

5   considerable overlap between the snowy and ridged categories (Fig. 13). However, the ridged category is quite dispersed, and may even consist of different classes of deformation which should not be grouped all together. Nevertheless, it is apparent that at the very least, the level and non-level categories are meaningfully distinguished. With more data and larger scan sizes (e.g. from OIB), a deep learning neural network suitable for unsupervised clustering (e.g. an autoencoder) could identify natural clusterings with their associated features (Baldi, 2012).

10   To emphasize the importance of the mean elevation, we also tried training the same ConvNet architecture with demeaned elevation as the input. Our ConvNet architecture is able to achieve a lowest validation error of 25% (training error 10%), but test MRE is relatively high (40%). The validation error is only slightly lower than the fit error for fitting $T \propto \sigma$ (Fig. 6b, with MRE 33%), and the test error is worse than the linear model, and has twice the test MRE of our ConvNet with the raw surface





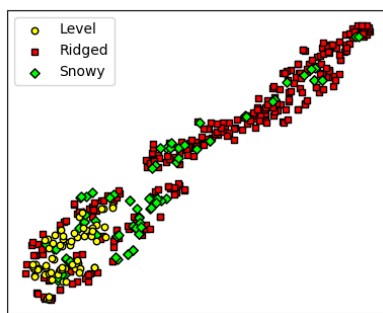

**Figure 13.** The t-SNE diagram for the encoded input, using the first fully-connected layer (feature vector of size 64) (Maaten and Hinton, 2008). The level and ridged categories are most clearly clustered, although the snowy category may also be a cluster. There is some overlap between the snowy/ridged clusters, which may reflect how ridges are often alongside snow features. It is also possible that the ridged category contains multiple different clusters. This result suggests that the manually-determined surface categories shown in Figs. 8 and 12 are pertinent, but perhaps not the most relevant, for estimating SIT given different surface conditions.

elevation (test MRE: 20%). Moreover, a simple statistical model of thickness as a linear function of surface elevation, snow depth and RMS roughness also only does marginally better (MRE 30%). This suggests both that the surface elevation means are important, and also that these means are differently treated for different features, as was speculated in the previous paragraph.

We also trained the ConvNet to predict the mean snow depth, with comparable training/validation/test errors of 15%/17%/18% when using raw lidar input, and errors of 15%/22%/45% when using demeaned lidar input, which suggests the same analyses hold for snow depth prediction. As the snow depth is largely correlated with the surface elevation, with the exception of ridged areas, it is not surprising that the demeaned input is not a good predictor of the snow depth. However, when metrics obtained from the demeaned elevation (such as roughness) are combined with the mean elevation, snow depth estimates (as well as SIT estimates) are improved. This may mean that both SIT and snow depth may be predicted given some lidar input, which is promising for applications to larger datasets such as OIB or ICESat-2.

Interestingly, predicting mean draft thicknesses using demeaned AUV windows gave low errors of 9%/11%/14%, suggesting that ice-surface morphology is a far better predictor of thickness than snow-surface morphology. This is not surprising, given that snow obscures the deformed ice surface. To compensate, the mean elevation becomes much more important.

Another approach to analyze these learned weights is to look at the sign of the weight and the typical values of the activations in Fig. 12. Feature #0 has a negative weight for which the ridged category (and to a lesser extent, snowy) has the largest (most negative) feature values; this leads to adding extra thickness, primarily for the ridged ice category. This perhaps accounts for a higher percentage of ice freeboard in the surface elevation measurement than for the level and snowy categories. Indeed, most of the level category have values near 0 for this feature. This could therefore be interpreted as a 'deformation correction' of some sort, or increasing the effective density of the ridged surface (perhaps due to a higher proportion of ice). This is also the case for features #3 and #6, which is not surprising as these three features all had strong correlations to the mean elevation for the ridged/snowy categories.



Feature #7, which has positively- and negatively-skewed distributions for level/ridged categories respectively, centered on the snowy category distribution, may be accounting for variations in snow density. For example, ridges may have less wind-packed snow due to the shielding effects of the ridge (and hence less dense snow), whereas level surfaces may have wind-packed, denser snow. In contrast, feature #5 has the level/ridged distributions skewed the other way around. Because the weight is positive, but the values are mostly negative, this most strongly reduces the thickness estimate for level surfaces. This may be equivalent to reducing the effective density of the surface due to the presence of snow, which would be reduced the most for level surfaces (that have mostly snow), whereas the ridged category would have a minor correction (and so the feature values are mostly near 0).

The inner workings of ConvNets are not easily interpreted, but the analysis here suggests that the ConvNet is responding in physically realistic ways to the surface morphology. It may be possible to use these physical metrics to construct an analytical approximation to the model, but due to the nonlinearities in the ConvNet as well as the considerable scatter between the features and our guessed metrics, this will not be as accurate as simply passing the input through the ConvNet.

In applying any model to a new dataset, it is assumed that the relationships from the fitted dataset hold for the new dataset. We already showed that linear fits do not hold for different datasets (even from the same region/season), with the relative error increasing by factors of 2-4 when estimating local or floe-wide thicknesses, likely due to differing snow/ice proportions in the surface elevation. This is true even when applying relationships from some PIPERS floes on other PIPERS floes. In addition to different surveys having different freeboards, ice/snow densities may also be differently distributed between surveys. Our ConvNet has errors of 12-20% when estimating both the local and scan-wide thicknesses of a new dataset, which is only slightly higher than the validation errors of 7-15%. This suggests that the morphological relationships learned in the ConvNet also hold for other floes of comparable climatology, which in turn suggests that deformation morphology may be consistent within the same region/season.

Although our survey consists of high-resolution lidar, snow and AUV data, we really only need high-resolution lidar data. We showed that using demeaned AUV topography has the same low error in predicting mean thickness as using the surface elevation. However, lidar surveys are much easier to conduct, and so a more viable method for obtaining more data for future studies is to use a high-resolution lidar scan, combined with coarser measurements of mean sea ice thickness (e.g. with electro-magnetic methods, as in Haas (1998)). Snow depth measurements are not needed with this method. This should greatly reduce the logistical difficulties to extend these methods to more regions/seasons.

## 5 Conclusions

Statistical models for SIT estimation suffer from a lack of generalization when applied to new datasets, leading to high relative errors of up to 50%. This is problematic if attempting to detect interannual variability or trends in ice thickness for a region. Deep learning techniques offer considerably improved accuracy and generalization in estimating Antarctic sea ice thickness. Our ConvNet has comparable accuracy to a linear fit (16% MRE vs. 20% MRE for fitting PIP4-8), but it has much better generalization to an unseen floe (20% MRE vs. 28% MRE for applying the best linear fit). This linear fit uses both an unphysical



constant term, as well as snow depth data that is not needed for the ConvNet. If comparing to a linear fit with no constant and without snow depth data, then the linear fit has a far higher fit error (43% MRE) and far worse generalization (47% MRES) than the ConvNet. The low test error for the ConvNet suggests that surface morphology, as identified by the ConvNet, may be consistent between different floes from the same climatology, and that this morphology may inform estimates of SIT.

Another strength of our proposed ConvNet is that it can account for a varying ice/snow density, with greater complexity and accuracy than an empirical, regime-based method. Although recent works like Li et al. (2018) have attempted to vary effective surface densities using empirical fits, these are not effective at higher resolutions, where snow/ice proportions may vary locally. Although the workings of ConvNets are somewhat opaque, we have shown that our ConvNet takes into account the spatial structures of the deformation, and given plausible justifications for why the snowy, level and ridged surfaces are
treated differently. The learned filters suggest that morphological elements are important for SIT estimation. We find that even for level surfaces, there is a considerable varying ice freeboard component that creates an irreducible error in simple statistical models, but can be accommodated as a morphological feature in a ConvNet. Our error in estimating the local SIT is <20% (RMS error of ∼7 cm), which is considerably lower and at higher resolution than current satellite-based estimates (∼50%, or 80 cm, see Kern and Spreen (2015)), and the resulting mean scan-wide SIT also has lower errors (RMS error: 2-3 cm) than
empirical methods (16 cm, see Ozsoy-Cicek et al. (2013)).

Although our ConvNet would be greatly improved with more training data, it is promising that local sea ice thickness can be accurately predicted given only surface elevation measurements. More extensive lidar/AUV/snow measurements from different regions/seasons would improve the ConvNet generalization, but because high-resolution ice thickness and snow depth are not needed, other, simpler-to-obtain data sets (e.g. coincident scanning lidar and EM-induction ice thickness measurements) can
also be used with this technique. The window size of 20 m x 20 m used here may also be valid, with some modifications, to work on OIB lidar data, as the learned features at ∼8 m resolution are also resolved by OIB lidar data (resolution 1-3 m). Using a larger training set, it may be possible to more readily identify relevant metrics for predicting SIT that may be measured/inferred from low-resolution, coarser data like ICESat-2. With more data, the low errors of deep learning-based methods may yield high-resolution, low-error SIT estimates that may be able to verify modest interannual variability.

*Competing interests.* The authors declare they have no competing interests.

*Author contributions.* JM and TM conceived of the research idea and collected field data. The manuscript was written by JM
and edited by TM. HS wrote the code for processing the AUV data and JM wrote the code for analyzing the data.

*Data availability.* The entire PIPERS dataset is expected to be uploaded to a publicly-accessible portal in late 2019. However, all the layer cake data used here will be uploaded upon publication of this paper.





*Acknowledgements.* This work was supported by U.S. National Science Foundation grants ANT-1341606, ANT-1142075 and the U.S. Office of Naval Research N00014-13-1-0434. The authors would like to thank Blake Weissling for providing the lidar data, and Jeff Anderson for technical support in the AUV missions. Guy Williams and Alek Razdan were instrumental for collecting the AUV data. The crew onboard the RV N. B. Palmer were also essential to the success of the PIPERS expedition.

## 5 Appendix A: ConvNet Details

For a comprehensive introduction to deep learning, the reader is directed to Shalev-Shwartz and Ben-David (2014). Here we will give the details of our ConvNet and explain the importance of chosen parameters.

Convolutional Neural Networks, commonly known as ConvNets, are a class of deep neural networks that convolve filters (matrices that contain weighting coefficients, or weights) through the input array. The input array is typically an image, and the learned filters typically correspond to basic edge detections in initial layers, and more complex features in later layers (e.g. Krizhevsky et al., 2012). Here, we use the lidar elevation scan as an input, due to its similarity to a grayscale image.

Like other deep learning methods, ConvNets 'learn' by updating their weights. This is done through comparing the output of the prediction with the true output, using the derivative of a loss function (here, mean squared error) propagated through the layers in reverse (backpropagation). The weight update rule, in its most basic form, is $w_{i+1} = w_i + \eta \frac{\partial E}{\partial w_i}$, for some weight $w$, loss function $E$ and learning rate $\eta$. The value of $\eta$ is important to ensure convergence: too high, and the filters may not converge (and may even diverge); too low, and the filters may take too long to converge. In order to introduce nonlinearities in the network, a nonlinear activation function is used at each layer. Typically, this is done with a Rectified Linear Unit (ReLU), which zeros out all negative activations. We chose a scaled exponential linear unit (SELU), which has been found to improve convergence (Klambauer et al., 2017), as ReLUs sometimes lead to dead weights when dealing with many negative values. As convolutions by default shift by 1 pixel at a time, this leads to considerable overlap and large output sizes at each layer. To combat this, the filters can shift by a different number; this is called the stride.

ConvNets are normally used in image classification problems due to their ability to discern features. The output would be a probability vector assigining likelihood of different classes, with the highest one being the prediction. ConvNets can also be applied to regression problems (e.g. Levi and Hassner, 2015) by simply changing the output to be one number. Here, we make the output the mean thickness, scaled by 5. The scaling here is because, for our dataset, the maximum thickness was just under 5.0 m, and normalizing the outputs to be between 0-1 allows the gradients for the backpropagation of error to neither vanish nor blow up. Similarly, the lidar inputs were scaled by 2.0 to keep them between 0-1. The values are unscaled during model evaluation. ConvNet inputs, when dealing with image classification, are often standardized to have a mean of 0 and a variance of 1, but this was not done here as we want to use the mean and variance (roughness) of the elevation to predict the mean ice thickness.

The proposed architecture is shown in Fig. 14. We use multiple convolutional layers to try to capture morphological features, along with fully connected layers at the end to combine the learned features. We tried networks with 2, 3 and 4 convolutional layers and 1 or 2 fully connected layers with a variety of filter sizes and found the one shown in Fig. 14, with a total of 5 hidden




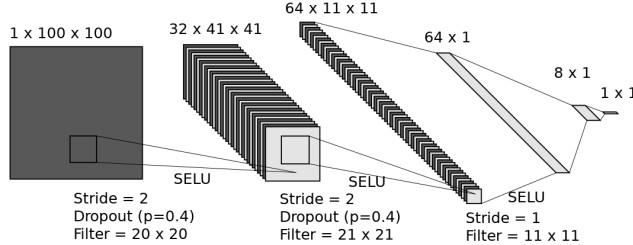

**Figure 14.** ConvNet architecture, using 3 convolutional layers and 2 fully-connected layers, for predicting the mean thickness (1 x 1 output) of a 20 m x 20 m (100 x 100 input) lidar scan window at 0.2 m resolution (LeNail, 2019). The (64 x 1) layer is made by reshaping the (64 x 1 x 1) output of the final convolutional layer, and so is visually combined into one layer. The optimzer used was Adam with weight decay $1.0 \times 10^{-5}$ (Kingma and Ba, 2014). The initial learning rate was $\eta = 3 \times 10^{-3}$ and reduced by a factor of 0.3 every 100 epochs until it reached $9 \times 10^{-5}$.

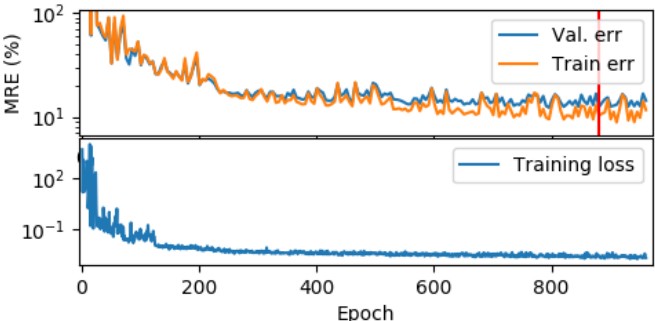

**Figure 15.** Training errors, validation errors and training losses shown on a logarithmic scale. Although the training loss continues to slowly drop after the epoch with the lowest validation error (red line, at epoch 881), validation error stays relatively flat, suggesting that the ConvNet is overfitting after this epoch.

layers, had the best results. The filter sizes were chosen to try and capture feature sizes of <20 m, following Section 3.1.2. The first layer has a size of 4 m, the second is 8.4 m, and the third is 8.8 m. For the first two layers, a stride of 2 was used to reduce the dimensionality of the data. The implementation was done using PyTorch with an NVIDIA Quadro K620 GPU.

The input windows were randomly flipped and rotated in integer multiples of $90^o$ to help improve model generalization.
5   Dropout, which randomly deactivates certain weights with some probability $p$, were added after the first and second convolutional layers ($p = 0.4$) to reduce overfitting (Srivastava et al., 2014). The selected model for analysis was the best-performing validation error (15.5%) at epoch 881, as shown in Fig. 15.



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
