# Peer review of "Estimating Early-Winter Antarctic Sea Ice Thickness From Deformed Ice Morphology"

_The Cryosphere, 2019_

## Referee Comment (RC1) · Anonymous Referee #1 · 7 Aug 2019

**General comments**

This manuscript introduces estimating deformed sea ice thickness with in-situ data using simple statistical methods and deep learning technique. Although I believe Convolutional Neural Network (CNN) can be an alternative way to retrieve sea ice thickness without snow depth and densities, the readability of this manuscript is low. I think major revisions are needed before publishing.

**Specific comments**

Introduction: the long introduction distracts the objective of this manuscript. The authors should concise previous literatures in the introduction. The authors should focus on more the objective of this study in the introduction.

P2 L27: surface elevation normally means surface height with respect to Earth ellipsoid in the altimetric study. Surface elevation and freeboard are used the same meaning in this manuscript, which can confuse the reader. I would suggest change surface elevation to freeboard throughout the manuscript.

P2 L29: since the hydrostatic equilibrium equation depends on altimeter type (i.e., laser/radar) it would be good to mention which one.

P7: This manuscript majorly covers the methodology for estimating sea ice thickness. I believe this manuscript should include method section in the main body for better understanding to readers. This manuscript needs a method encompasses the entire manuscript. Particularly, as CNN is rather highlighting in this manuscript CNN details should be in the body manuscript.

Table 1: Please briefly explain in terms of sail angle.

P9 L7: There is no validation for the model in 3.1.1 and 3.1.2, which is not consistent throughout the manuscript. Do authors have a specific meaning without the validation?

P12 L15: Please briefly explain in terms of drill lines.

P13 L17: Why this particular range? (2.9-6.1)

P13 L23: What is the basis for setting 5.9?

P14 Figure 7: While freeboard (F) is mentioned in figure 7, surface elevation (F) is mentioned in the caption, which is not consistent.

P14 L8-17: this paragraph should be in the discussion.

P16 Table 2: Please briefly explain in terms of Akaike Information Criterion (AIC).

P17 Table 3: Why the authors separate linear model (i.e., without constant vs. with constant).

P17 L10-L18 - P18 L1-12: this part should be in the discussion.

P18 L13: It would be better the authors include the spatial distribution of sea ice thickness derived by CNN with discussion.

P18: the first paragraph of 3.3 should be in the methods.

P19 L12: Normally this parameter setting is determined by trial and error.

P20 Figure 9: Some part of the caption of figure 9 should be in the main body. (from we also to the end).

P18-21: 3.3 predicting SIT with deep learning is quite mixed with methods, results, and discussion. please reorganize 3.3.

P25 L15: what is meant by Figure #0?

P26 L31: as the validation of this method is spatially limited, this sentence should be corrected.

**Technical corrections**

P2 L9: wieth -> with

P7 L6: With -> with

P7 L7: need references.

P9 L11: I don't see ratio of keel depth and snow-sail height in the Table 1.

P17 Table: replace "no constant" with "without constant".

P19 L16: replace "CNN" with ConvNet.

P22 Figure 10: Figure 10 never mentioned before.

---

## Referee Comment (RC2) · Stefan Kern (Referee) · 13 Aug 2019

Review of

Estimating Early-Winter Antarctic sea ice thickness from deformed ice morphology

by

Mei, J. M., et al.

Summary: A suite of very-high resolution contemporary sea-ice draft, snow depth, and surface elevation measurements carried out during the PIPERS expedition into the Ross Sea in May/June 2017 is analysed. First, it is used to obtain a set of very-high resolution sea-ice thickness distribution for the probed sites; this is used as a benchmark data set for the rest of the paper. Secondly it is used to investigate ratios and relationships between the three measured and the one computed (sea-ice thickness) parameter paying special attention to a discrimination between level and deformed areas. Various linear models are subsequently applied (and compared to the literature) to figure out which of the parameters measured need to be used and/or combined to obtain an optimal sea-ice thickness product - compared to the above benchmack data set. An attempt is made to relate variations in the coefficients used in the linear models to variations in effective density. Finally, based on these results, a deep learning convolutional neural network is fed with surface elevation information, trained, evaluated and applied to predict sea-ice thickness distributions of independent very-high resolution PIPERS elevation data sets - with quite convincing results with respect to the mean relative error. It is shown that the application of the network trained with PIPERS data is not necessarily succesful in predicting sea-ice thickness from surface elevation measurements of a different expedition.

I found the paper easy to read.

I definitely support publication in "The Cryosphere".

Despite the current quality of the paper I recommend that the authors invest some effort to better explain a few issues, to also improve one or two of the figures, and to better structure it by means of creating a separate methodologies section. I am inclined to rate this a major revision. The main concerns I list in the general comments (following immediately below), whereas the smaller - often less general issues - I noted in the specific comments. You will also find some typos noted.

General Comments: GC1: I note that a dedicated "Methods" section is missing completely. It is Data followed by Results. There are places where this seems ok for the flow of the manuscript but there are other places, e.g. Section 3.2 where this seems not to be optimal. The deeper I stepped into this section the more confused I got. At a certain point I got lost with density values and with regressions with or without intercepts or additional constants. This section would perhaps benefit from a clear up-front explanation of what you did / how you derived coefficients / which density values you choose (and why) / how you derive effective density values (and why)? Such an improvement in structure of the paper would possibly also reduce its length a bit here and there.

GC2: You treat the hydrostatic equation as a form of a linear fit. While one can see this as such a fit it would be very important to mention (even more) that the coefficients as you call them are based on density values and are computed based on physics. This is an important difference to the empirical linear fits used by Xie et al. or Ozsoy-Cicek et al. which are purely mathematical. To my opinion it would add to the understanding of your paper if you would clarify this even better at an appropriate position in your paper. I'd think that interpreting the CNN results into the direction that effective densities can be derived is very hypothetical - especially given the unknown (and non-existing) relationship between sea-ice and snow densities which are both involved. I note in this context that the issue of negative ice freeboards has neither been mentioned nor discussed. I guess it would not hurt to get back to it given the results published in Ozsoy-Cicek et al. (2013) and Yi et al. (2011).

GC3: I am missing the presentation / discussion of more results of the ConvNet approach. What a reader might have loved to see is profiles of sea-ice thickness computed from the draft-snow depth-surface elevation measurements (your benchmark) and of the sea-ice thickness estimated with your approach. Ideally you are able to show at least one representative profile of each PIP used here. That way one will get a better handle on the actually estimated sea-ice thickness distribution compared to the measured one - in addition to the histograms shown.

GC4: ICESat-2 is up since September last year. After having read the paper I am wondering what the ultimate goal of your work is. Is it to create high-resolution validation data sets of the sea-ice thickness which are spatially distributed? Or is it to develop an algorithm which potentially could be applied to ICESat-2 data. For both cases, I believe

the authors could stress the main motivation and future use of their work and product.

Specific Comments:

Page 2, Line 14: I suggest to cite the paper by Behrendt et al., 2013, Sea ice draft in the Weddell Sea, Earth System Science Data, 5, measured by upward looking sonarsto underline that also this ULS data is a valuable source - even though you write "sporadic"

Page 3, Line 1-9: I guess it would not hurt to perhaps again refer back to Kern and Spreen (2015) who dedicated some work on the uncertainty analysis for ICESat sea-ice thickness retrieval and to Kern et al., 2016, Antarctic sea-ice thickness retrieval from ICESat: Inter-comparison of different approaches, Remote Sensing, 8(7), who inter-compared a number of sea-ice thickness retrieval approaches for the Antarctic and on which results the Li et al. (2018) paper cited is based upon. I am tempted to say that the fits used by Li et al. (2018) are based on the work of Ozsoy-Cicek et al. solely and not on the work of Xie et al. (2011).

Line 14: "do not yet understand the distribution" –> I am inclined to say that we do understand the physical mechanisms forcing the snow depth distribution around ridges very well. What we cannot yet do is, however, to measure this distribution accurately over an large enough area.

Line 10-19: In this paragraph, the work of Weissling and Ackley, 2011, Antarctic sea-ice altimetry: scale and resolution effects on derived ice thickness distribution, Ann. Glaciol. 52(57) might fit as well.

Lines 16/18: This is perhaps a good place to refer to the work of Hutchings et al., 2015, Comparing methods of measuring sea ice density in the East Antarctic, Ann. Glaciol., 56(69)

Lines 20-22: I agree that the unknown snow depth is one factor here. But isn't the fact that we don't know the keel morphology and distribution relative to what we see from above with a LIDAR contributing much more to a potential bias in estimated sea-ice

thickness?

Line 23-27: As far as I know, Kern and Spreen (2015) focused quite a bit on ICESat and the uncertainties involved. I doubt, however, that this is the correct citation for the AMSR-E snow depth bias issue. I'd say the first to report this issue were Worby et al., 2008, Evaluation of AMSR-E snow depth product over East Antarctic sea ice using in situ measurements and aerial photography, J. Geophys. Res., 113. Their work was followed later by Ozsoy-Cicek et al., 2011, Intercomparison of Antarctic sea ice types from visual ship. RADARSAT-1 SAR, Envisat ASAR, QuikSCAT, and AMSR-E satellite observations in the Bellingshausen Sea, Ann. Glaciol., 52(57) or Kern et al., 2011, An intercomparison between AMSR-E snow depth and satellite C- and Ku-band radar backscatter data for Antarctic sea ice, Ann. Glaciol. 52(57). In the same Ann. Glaciol. volume you also find the paper by Markus et al., 2011, Freeboard, snow depth and sea-ice roughness in East Antarctica from in situ and multiple satellite data, Ann. Glaciol., 52(57). Another paper about the deficiencies of the AMSR-E snow depth product could be this one: Kern and Ozsoy-Cicek, 2016, Satellite Remote Sensing of snow depth on Antarctic sea ice: An inter-comparison of two empirical approaches, Remote Sensing, 8(6).

Line 33: "fewer such datasets exist" –> This applies to the Antarctic and I would mention this accordingly. In the Arctic there are way more draft measurements available and these have actually been used to develop draft-based sea-ice thickness estimation tools.

Figure 1: I love this figure. It could be even a tiny bit more realistic if the ice floe or sheet would not be continuous in the ridge / keel area.

Page 4: Lines 3-12: I am wondering whether in the context of this discussion the work of Goebell, 2011, Comparison of coincident snow-freeboard and sea ice thickness profiles derived from helicopter-borne laser altimetry and electromagnetic induction sounding, J. Geophys. Res., 116 should be mentioned as well?

Page 6: I suggest to mention / give answers to the following questions here: - Water depth in which the AUV was operating - How many AUV scans per "cake" were stitched together? - If multiples scans: Were all scans carried out into the same direction? Or parallel to each other in opposite directions? X-ing? - Did I understand correctly that per "cake" 4 surface elevation scans were carried out, each from one side of a 100 m x 100 m grid? Or did you actually fly over the area? - I assume the snow depth measurements were the last measurements carried out –> although it is logical it is worth to mention this. - It is not entirely clear how the about 2000 measurements per "cake" are distributed across the "cake" area and how this was technically realized. I assume that the measurements were carried out along parallel transects across the cake with a fixed transect-to-transect distance and that only the sampling along each single transect varies between 5 m and 0.1 m. - How is the reference sea-surface height computed and how accurate is it?

Line 29: "The ice thickness can ..." –> so there were no drillings?

Figure 5: Readability of this figure would improve with an increase of its size.

Page 11:

Line 18: "very similar value of 1.3" –> This very similar value needs two standard deviations (2 time 0.1) to include that 1.5 values from other studies. Perhaps "similar" would do it?

Figure 6: - See comment to Figure 5. - In the caption I would call the black line dashed rather than dotted.

Page 12:

Line 11: "The lidar and AUV data were corrected by ..." –> I don't understand what needs to be corrected here. Is there a way you specify better what you did? Why is this a "correction"?

Page 13: Line 12: "snow = surface elevation assumption" –> I recommend to mention

that this is a strong assumption, that it applies to thin, perhaps medium thick first-year ice only (possibly only to the kind grown under quiescent conditions, i.e. not originating from the pancake-ice cycle), that it requires a certain snow load to be present, and that such an assumption can only be made if one is not interested in a really exact sea-ice thickness estimate.

Line 17: "All our coefficients" –> Would you mind to refer to the place where you already mentioned these coefficients?

Line 21: "2.2-3.1 in Ozsoy-Cicek et al (2013)" –> I tried to figure out how you ended up with this range and potentially misunderstood something. If I check that paper, then - in Figure 5, which is possibly the one you got these numbers from, - I find regression lines with a considerable intercept between ~10 cm and ~30 cm, depending on the region, paired with this range in factor of F of 2.2 - 3.1. But these values are valid for positive ice freeboards only. When taking all ice freeboards into account, then the black lines (and numbers) in that Figure 5 apply. In addition to that: I could make sense to focus only on the Ross Sea results from that paper?

Lines 30-32: "This means that assuming ..." –> So what you state here basically is, that the linear regression approaches developed by Xie et al. and Ozsoy-Cicek et al. are of limited value? If so you could mention this and also refer to Kern et al. (2016) in Remote Sensing, where it is layed out that the linear regression approaches fail to provide a meaningful circum-Antarctic sea-ice thickness distribution.

Figure 7 - Some data points are annotated "snowy" –> I did not find an explanation of what this is in the text or in the caption. Where is the distinction between "snowy" and "ridged"? - caption: The ice density value given in line 4 of the caption differs from the one given in the text on page 13, line 25.

Page 14: Line 3: "T = 2.45F + 0.21" –> is modified from Ozsoy-Cicek et al., now using unit meters instead of centimetres, correct? "for a winter Ross Sea" –> according to Ozsoy-Cicek et al. (2013) this is data from just one cruise in Sep./Oct. = much later in

the season than PIPERS. In that sense your statement in Line 6 "same region/season" should perhaps be changed? Also the spatial overlap (see Ozsoy-Cicek et al., 2013, Figure 1) is quite small.

Line 4: So your intercept is -0.73 meters or -73 cm? That is quite large.

Line 7: "nonzero freeboard" –> "nonzero ice freeboard"

Lines 15-17: Yes, I agree with your interpretation. However, it might make sense to also mention that the Ross Sea data used in Ozsoy-Cicek et al. (2013) was from a different part of the Ross Sea and from a different season and to my opinion indeed exhibits a totally different characteristics than the PIPERS data set collected 3-4 months earlier.

Page 15: Line 6: "additive constant" –> I don't understand what you mean by this. Did you add an intercept?

Lines 7 & 8: Isn't it surprizing that the coefficient for F fitted over all four PIPs of 10.4 is so close at the upper range of 10.6 for individual PIP fitting? Also: The range for the coefficient for D of the individual PIPs does not include the value found over all four PIPs. Is this logical?

Lines 9-12: - Your measured snow densities are considerably lower than those given by Sturm et al. (1998). Could it be that the latter were obtained in late winter / spring? - While I understand the concept behind the effective sea-ice density (voids filled with water included in the density estimate) I have problems to understand the concept of an effective snow density. What is this? In this context, I find your effective snow density value to be quite high. - I guess it would be good to learn how you ended up with the density values reported in Line 10. When I tried to insert your range for the factor for F (7.9 to 10.6) into an equation where D is zero, then I end up with densities between 897.9 and 931.0 kg/m^3. But of course, without further information from your side I cannot reproduce your numbers. - I find it quite surprizing that the standard error for the effective sea-ice density is so low compared to that of the effective snow density.

- You use a water density value which differs from those given at the beginning of this paragraph. Why? Where does this value originate from?

Lines 15-18: Please check these sentences. There is some repetition first and then something is missing.

Lines 20-22: "For example, ..." –> Just to understand this: What you write here in the text is the comparison between using coefficients of ONE of the PIPs to estimate sea-ice thickness in another PIP while in Table 3 you show the comparison between using a joint coeffient of THREE PIPs to estimate sea-ice thickness in the remaining PIP. I just got confused a bit about why you write different things in the text than you actually show in Table 3 (and refer to in the subsequent sentence).

Table 2: - "no int." means what? - Are you sure the AIC is a monotonic function even on the negative value range? I am just wondering whether the "smallest" AIC criterion does not need to be applied to absolute values? Could it be that these negative values have no proper meaning in case that the correlations are so low? - The subscript "adj" stands for "adjunct"? "adjusted"? If the latter adjusted to what? - What is the unit of the constant? It seems to be in meters?

Page 17: Table 3, caption, last line: "zero freeboard = zero thickness condition"? Do you refer to ice freeboard here? Do you perhaps mean "zero snow depth"?

Line 1: This equation is something you could use in a "Methods" section (should you include one) to tackle my general comment GC1. I note however, that seemingly with this equation one can explain only parts of the entries in Table 2; the c3 times sigma part is not represented in Table 2.

Lines 5-9: It is still not clear to me how you discriminate between "snowy" and "level". Please add.

Page 18: Lines 3-6: I can in principle follow your argumentation that zero surface elevation (= zero snow depth) means zero sea-ice thickness. I would sign this if we

consider larger scales. But on the scales investigated with the PIPs this is not necessarily true because under cold conditions and hence impermeable sea ice there will be many places with a negative ice freeboard. Even if we assume for simplicity that most of these will have a snow cover and hence potentially have a non-zero surface elevation, it is still likely that especially in the vicinity of ridges and/or where the ice is under lateral stress - at the scales of your measurements - you will have surface elevations close to zero or even negative ones paired with a non-zero sea-ice thickness. - This paragraph is again a good place to comment and/or underline the difference between the physically based coefficients used by Zwally et al. and similar papers and the empirically based coefficients used by Xie et al and similar papers. One could argue that the physically-based coefficients are more dependent on the validity of the hydrostatic assumption while the empirically based ones are not ... but I am not sure this holds.

Page 19: Line 13/14: "20% of the data ..." –> this refers to the randomly selected data? If so, please stress so in the text.

Line 13 vs. Line 16 and remainder of the text: Please check your usage of "floe". From the text until here I got the impression that the PIPs are subsets of one floe. Here I get the impression that PIPs comprise several floes out of which a few are selected. Please clarify your terminology here.

Line 20: You state PIP8 here but in Figure 9 it seems you refer to PIP9. Please check.

Line 24: "epoch 881" –> does it make sense to refer to the Appendix here? Otherwise this information is perhaps a bit out of context.

Line 31: "are all negative" –> except for level ice.

Line 33-35 and beyond: I doubt that this comparison should be presented as is. Aren't these data sets quite different? I wrote about the sub-set of data for the Ross Sea used in Ozsoy-Cicek et al. (2013) already. In Li et al. (2018), the data basis is ICESat footprint-scale estimates of the freeboard - hence we talk about one value for one

footprint of which we do not know how well it covers how many different surfaces. The data set used in Ozsoy-Cicek et al. (2013) is at least based on multiple measurements conducted on one or more transects across a single floe.

Figure 9: The description / caption of the figure needs to be improved. - Please annotate the images with a), b), c) ... - What is the value behind showing a continuous fit in addition to the bars? It extrapolates the bars towards non-existing data values. - What are the bin-sizes used? Are these the same in all three histograms shown? Do they always have the same borders (i.e. minimum and maximum value included into the count of a respectiv bar)? - The peak counts are obscured by the legend. This needs to be changed. - I suggest to add in each row which PIPs are used for what. - I suggest to stress in the caption that the last row shows a different range of thickness values. - The caption in line 3 says PIP7-9 but in line 2 it is PIP4, 7 and 8. What is correct? - The caption in line 4 says PIP9. True? - What is the unit at the y-axis in the histograms? - The mixed colors in the histograms originating from overlapping bars of different data sets are not easy to interpret. Perhaps you could either add these in the annotation (which is possible if you increase the size of Figure 9) or find a different way to show the counts of the different data sets. One way to do this would be that you use substantially narrower bars which you do not let overlap each other and center these at a specific thickness; then in the caption you might need to state that you display three bars centered at a specific thickness, separated horizontally for better visibility. That way the real differences in the distributions would become more clear.

Page 20: Lines 2-4: Perhaps you could put these numbers in context with the number of data points used to get these uncertainty estimates? I guess, in case of Ozsoy-Cicek et al. (2013) we are talking about 23 floes with an actually unknown number of measurements per transect. About how many measurements are we talking in your case?

Page 21: Line 14: I suggest to remove the "see" –> at least I cannot see these results.

Line 29: "isostatic assumption may no longer be valid" –> may be so. What do we know about spatial scales over which the isostatic assumption is valid? No too much I'd say - particularly for ridged ice. Perhaps this sentence could be deleted.

Figure 10: - this figure belongs to section 4 and should be located within section 4 not before it. - Why do we have 3x3 imagettes for layer 1 but 16 for layer 3? - The size of 4 m and 8.8 m given in the caption, do these refer to the pixel size in these imagettes or to the imagette size itself? It seems as if the pixels in layer 1 are indeed smaller than in layer 3. - Instead of "as the lidar" you might want to write "as the surface elevation" - Is it in this context correct to assume that layer 3 has the unit meters while layer 1 is unitless? - What do the bright and dark pixels in layer 1 mean?

Page 22: Line 17 through Page 23: Line 7 and Figure 11: - Please provide a), b), ... in Figure 11; it aids referring to the images. - I suggest to mention Figure 11 before Figure 12. - You refer to Figure 11 in Line 5 but should perhaps also do it in Line 3 (strong correlation for feature #0) and again in Line 6. - I have difficulties to understand the continued mentioning of "effective densities". I doubt that with the CNN you can (and should) derive any conclusions about the effective density - especially because the densities for sea ice and snow do not necessarily co-vary. This brings be back to GC2.

Page 23: Line 10: "snowy surfaces" –> which still need to be defined in comparison to "level surfaces".

Page 24: Line 4: "ridged and level surfaces are clearly distinguishable" –> I don't agree when I look at Figure 13 - unless I have perhaps misunderstood what the used tool is able to show. But my interpretation of this figure is that level, ridged and snowy symbols overlap well.

Page 25: Lines 9/10: That prediction of snow depth from lidar input is possible as also been shown by Ozsoy-Cicek et al. (2013) and Kern and Ozsoy-Cicek (2016).

Lines 11-13: I guess these two sentences could be deleted.

Page 26:

Lines 1-8: I am not a fan of these attempts to try to relate CNN features to (effective) snow density variations which may or may not be realistic and physically meaningfully linked to input parameters. To my opinion, this really requires a careful analysis and description of how the CNN "learns" from the input data and whether there is (within the CNN) a link to physics - which I doubt is the case.

Lines 18: "thickness of a new dataset" –> you seem to have applied your approach to a different PIPERS data set. It might be really beneficial to show this example in the paper and not to just mention it. Particularly because you come back to this in your conclusions (Line 33, "unseen floe").

Page 27: Line 5: "it can account for a varying ice/snow density" –> I'd say that this is a hypothesis. It may be that the ConvNet is able to account for the different densities and perhaps even provide additional information about these - but the evaluation of whether this is the case and/or whether this is at all meaningful physically based on deep learning is not known and might not be over-stressed here.

Lines 12-15: "Our error ..." I suggest to not overstress these inter-comparisons because these are based on completely different data sets and scales. After all, a real quality measure of your method will be its application to ICESat-2 data which should be the overall goal here - as is finally mentioned in the last paragraph.

References: You need to go through the references list and complete it with respect to page numbers and journal volume and issue numbers. Also doi's are generally missing. Some journal abbreviations are not in place.

Page 28: Line 31: I am not sure but I guess Figures in the Appendix need to be named differently to the main text. See the instructions for authors.

Page 29: Line 2: Would you mind commenting on the layer sizes being first 4 m, then

8.4 m and subsequently 8.8 m? Do these "strange" values have to do with the pixel size of 0.2 m?

Figure 15: - What is a "training loss"? - It appears that after epoch ∼550 there is a small jump in validation and training error from a certain level before that epoch to a certain, lower level afterwards. Any explanation to this? - What explains the sudden increases in the training error from a low background of ∼15-16% MRE to the level of the validation error? It seems as if the result of the ConvNet even after that many epochs is still not stable?

Typos:

Please replace "e.g" by "e.g." (a few incidences) Please check usage of "climatology" and replace all incidences in the paper by a more appropriate term.

Page 2: Line 9: witeh –> with, interrannual –> interannual

Page 9: Line 26/27: "beyond beyond" –> "beyond"

Page 15: Line 11: "which" –> "who"

Page 21: Line 28: "slighly" –> "slightly"

Page 28: Line 23: "assigining" –> "assigning"

Figure 14, caption, line 3: "optimzer" –> "optimizer"

---

## Referee Comment (RC3) · Anonymous Referee #3 · 22 Aug 2019

Dear TC editor and authors of the manuscript TC-2019-140,

The topic of the manuscript is interesting and the content useful for sea ice research. A neural network has been applied to sea ice thickness (SIT) estimation from lidar surface elevation. The introduction section is quite comprehensive. It is mentioned that it may be possible that the introduced method may be used to improve SIT estimation by lower resolution / larger footprint laser instruments (ICESat-2). The results have mostly been presented nicely and comprehensively.

The first referee already submitted quite comprehensive comments on the manuscript, and I'll just try to complement his comments. I agree with him that a major review is still required before publication.

[Figure]

Here are some comments trying to improve the manuscript:

General comments:

1) I agree with the reviewer 1 that more results of the DCNN approach could be included.

2) Also a more detailed technical description of the applied methods (DCNN) would be preferable, as already suggested by referee 1. This could be a Section of its own (not an Appendix). Also include the information of numbers of DCNN neurons used at each layer and how these numbers were selected.

3) Regarding e.g. icesat-2 data, it would be nice to have some experiments or at least approximation related to the effect of resolution to SIT estimation using the proposed method.

More detailed comments:

S. Introduction P5, L17-18 "... detailed snow depth measurement": Also include already here by which method the snow depth measurements were made (not in detail).

S. Data P6, L12 and L22: instruments are named, also include references to their technical specs, and also shortly write on the principle of the snow measuring device.

S. Data P 6-7: Division of the data sets used into training and test data sets (possibly also validation data set) could be clearly described in the data section already. Were the data sets the same for all the performed experiments? This seems to be described later in the deep learning section for the DCNN.

S 3.1.1 P9, L18-19: Rather say "...Thickness of the level ice (L) forming a sail and its sail height (S)..."

S 3.1.1 P10, L8 "...for estimating sea ice thickness,..." -> "...for estimating sea ice thickness T,...". Possibly You could use SIT for sea ice thickness throughout the manuscript?

S 3.1.2 title could be "...mean sea ice thickness..." or "...mean ice thickness..." or even "mean SIT".

P10, Fig. 5. Make the figure larger, difficult to read in the printed version. Its width could e.g. be approximately the column width.

P11 S 3.1.2 L22: Describe the use of semivariogram in more detail. Did You make any experiments by varying the window size also?

P12 Fig. 6: Same thing as for Fig. 5, make larger.

P14 Fig. 7: Same thing as for Fig. 5, make larger.

P18 Fig. 8: Make the figure larger or make the box frames wider for better visibility. Include a legend describing the classes instead of writing it in the caption.

S 3.3, P 19 L5: The best-performing linear regression result has been given here for comparison. Have You any idea, could better results have been achieved by using a nonlinear approach with the same inputs, e.g. a multilayer perceptron neural network with the same inputs (plus an additional constant/intercept input)? Or are the dependencies really linear?

P19 L12-13: "20m x20m windows", also give the window size in pixels here.

Did You study the effect of the resolution to the result by using down-sampled data? Any idea, how would this possibly affect the estimation result? Possibly You could then get average SIT over a larger area? This could give an idea of the applicability of the method to coarser resolution data.

Figs. 9, 11,12,1314 and 15: make bigger for better readability in the printed version.

App. A: did You also vary the number of neurons at each level and how did this affect to the results? How were these parameters selected? Does there exist any "rules of thumb" for selecting the parameters (e.g. numbers of neurons) for DCNN's as there exist for Multilayer Perceptrons (as a function of the number of inputs and outputs).

App. A: Also include execution times for the training and SIT estimation in the used hardware.

And yet one interesting aspect: As a researcher of microwave and optical EO imagery (over sea ice) I am also interested in possibilities of utilizing the existing imaging devices for SIT estimation. Typical high-resolution (HR) sensors covering a wide spatial area, such as HR SAR or optical/IR sensors, measure only the 2-D sea ice surface, not the elevation directly. However, it is possible to locate ice ridges and even estimate their sail width in HR EO imagery. There is some literature (e.g. Timco & Burden, 1997) relating the ridge parameters to each. However, I have not seen any good reference relating sail width (Ws) to sail height (Hs). This kind of relationship would be very useful for better estimating ice thickness from 2-D HR EO data. Could the authors comment on this topic i.e. how (well) the morphology could be derived/estimated from the available 2-D EO data/imagery and whether this relation could be utilized in SIT estimation? Possibly a deep neural network could be used after deriving some ridge parameters from 2-D HR sea ice data form SAR/optical/IR, or even just training a DCNN with the data directly. This would naturally require a good data set with a large number of (nearly) simultaneous SIT measurements (possibly made by another validated remote sensing method, such as laser scanning).

Sincerely,

---

## Author Comment (AC1) · 9 Sep 2019

**We thank all three reviewers for thorough and helpful reviews.  Although reviewers suggested major revisions, almost all suggestions focussed  on the presentation.  The reviewers' main commentsrequested a dedicated methods section, more description of the ConvNet method in the main text (as opposed to in the appendix), and more exposition of the ConvNet results.  We have modified the manuscript accordingly and reorganized the manuscript as follows:**

**(1) We have expanded the description of data processing methods in Section 2, and added dedicated methods section (Section 3) that describes briefly the linear fit approach, and an extensive description of the implementation of the ConvNet.  Much of the latter is information that has been moved from the appendix, which is now short. We have kept the data processing separate from the data analysis (ConvNet) methods, as the former describes processing of the data to provide accurate data, while the latter describes how the processed data are analyzed.**
**(2) We have moved much of the description of the ConvNet implementation from the appendix to the main text**
**(3) We have added an additional figure of ConvNet results. As suggested by review #1, this shows a 2D comparison of both the linear fit and ConvNet ice thickness predictions to the original 2D mapped data.**

**In addition to suggested changes, we have removed the section on relationships between level ice thickness and ridge keel depths, as this was a diversion from the main results comparing simple linear and 1-D theoretical models with the ConvNet for estimation of ice thickness from surface topography, and not really connected with the main results. We have shortened the introduction accordingly (as also suggested by reviewer #1) to keep it focussed on the main thrust of the paper.**

**All reviewer comments are addressed inline below.**

**REVIEWER #1**

General comments
This manuscript introduces estimating deformed sea ice thickness with in-situ data using simple
statistical methods and deep learning technique. Although I believe Convolutional Neural Network
(CNN) can be an alternative way to retrieve sea ice thickness without snow depth and densities,
the readability of this manuscript is low. I think major revisions are needed before publishing.

Specific comments
Introduction: the long introduction distracts the objective of this manuscript. The authors should
concise previous literatures in the introduction. The authors should focus on more the objective of
this study in the introduction.
**The introduction has been shortened As decribed above, we have removed much of the material relating to ridge morphology as that is not germane to the main results.**

P2 L27: surface elevation normally means surface height with respect to Earth ellipsoid in the
altimetric study. Surface elevation and freeboard are used the same meaning in this manuscript,
which can confuse the reader. I would suggest change surface elevation to freeboard throughout
the manuscript.
**We have changed surface elevation or snow elevation to "snow freeboard", and freeboard (when referring to ice freeboard) to "ice freeboard" throughout.**

P2 L29: since the hydrostatic equilibrium equation depends on altimeter type (i.e., laser/radar) it
would be good to mention which one.
**As presented here, this equation is used as the universal 1-D buoyancy equation, and is valid independent of measurement. While different forms are used for different altimeters, this is merely a rearrangement of the terms based on what is measured (snow freeboard, ice freeboard, or potentially some horizon in between)**

P7: This manuscript majorly covers the methodology for estimating sea ice thickness. I believe
this manuscript should include method section in the main body for better understanding to readers.
This manuscript needs a method encompasses the entire manuscript. Particularly, as CNN is rather
highlighting in this manuscript CNN details should be in the body manuscript.
**We have included a Methods section that includes both background for the linear fit analysis, and a more extensive description of the ConvNet method and implementation, which included moving much of the material from the**

**appendixSome of the technical ConvNet details are kept in the Appendix to not distract from the main points. A Note, that the separate "Data and Processing" section describes the methods for acquisition and processing of the data. We have expanded this description, but kept it separate because it applies to all the subsequent analysis.**

Table 1: Please briefly explain in terms of sail angle.
**Defined sail angle, and clarified with "and a range of slopes across the deformed surface are given"**

P9 L7: There is no validation for the model in 3.1.1 and 3.1.2, which is not consistent throughout the manuscript. Do authors have a specific meaning without the validation?
**These sections were intended to compare our observations with prior studies, and because the fits are poor they are not validated. However, since section 3.1.1 (comparison between level ice thickness and keel depth) was not very relevant to other results of the paper we have removed this section along with much of the background information on ridge morphology to keep the paper focussed.**

**Section 3.1.2 was intended to demonstrate potential relationships between surface roughness and ice thickness (as suggested by previous authors) to motivate the use of surface morphology to aid in ice thickness estimation.  This section has been shortened and moved to Section 3.2.3 where the incorporation of surface roughness is included in the linear fit analysis**

P12 L15: Please briefly explain in terms of drill lines.
**This section has been removed as it is distracting from the key results in the paper, since drilling data are not used in the analysis. A short statement on the relative accuracy of drilling data has been added to clarify the corrections to the AUV data in the Data and Processing section.**

P13 L17: Why this particular range? (2.9-6.1)
**This comes from Table 2. We have added a note to refer to Table 2.  This section has been modified and moved to the discussion as it seeks to explain the differences between our fits and those of prior authors.**

P13 L23: What is the basis for setting 5.9?
**Following Fig. 10 (in new manuscript, 7 in old), this is the best fit line. As above, this section has been modified for clarity and moved to discussion. We have added a note to Fig 10.**

P14 Figure 7: While freeboard (F) is mentioned in figure 7, surface elevation (F) is mentioned in the caption, which is not consistent.
**This has been changed to snow freeboard.**

P14 L8-17: this paragraph should be in the discussion.
**This has been moved to the Discussion.**

P16 Table 2: Please briefly explain in terms of Akaike Information Criterion (AIC).
**We have added a note that the AIC attempts to minimize information loss, and that we use the lowest AIC to perform model selection.**

P17 Table 3: Why the authors separate linear model (i.e., without constant vs. with constant).
**As noted in Stefan Kern's review, the without-constant fit is an attempt to match physical conditions of hydrostatic equilibrium (and permit estimation of effective densities), whereas the with-constant fits are empirical and attempt to minimize fit error. The reorganization of the text separating out methods, results, and discussion should make this more clear. We use the fits with constant as a basis for comparison of how our ConvNet improves upon linear fits, and we use the fits without constant to estimate ice/snow densities in our data.**

P17 L10-L18 - P18 L1-12: this part should be in the discussion.
**We have kept part of this in the Results as it is reporting errors for some particular fit. As such, it is a result and fits best there, but we have moved the discussion of this to Discussion.**

P18 L13: It would be better the authors include the spatial distribution of sea ice thickness derived by CNN with discussion.

**This has been added to the manuscript as a new figure which shows that the the ConvNet prediction matches the spatial variability better than the linear fit (see below in response to Reviewer #2) for the figure) While we have produced this plot for each floe, we are electing to include one example as all floes show qualitatively similar results.**

P18: the first paragraph of 3.3 should be in the methods.
**The structure has been reworked as described above**

P19 L12: Normally this parameter setting is determined by trial and error.
**There are several reasons for this choice. Our goal here is not to make the best possible network, but to make a good network that can be interpreted to physically justify why the network is working. Our method is predicated on the assumption that feature morphology is important to SIT prediction. Since our feature sizes are of similar scale, we felt this was a reasonable choice so that the ConvNet would learn features that are likely physically relevant to sea ice thickness variability. With our limited dataset, we cannot use too large a window as this would lead to too few unique samples; similarly, if we use too small of a window,relevant physical features would not be captured. We have also tested the network by halving the window size (which has the danger of not capturing relevant physical feature scales), and by decreasing resolution, with no significant effect on performance as described in the text. Because of the limited size of the dataset, we cannot test this dependence further.**

P20 Figure 9: Some part of the caption of figure 9 should be in the main body. (from we also to the end).
**Caption has been edited. We keep a reference to the linear fit as a description of what is in the figure.**

P18-21: 3.3 predicting SIT with deep learning is quite mixed with methods, results, and discussion. please reorganize 3.3.
**We have reorganized as described above**

P25 L15: what is meant by Figure #0?
**The text reads "Feature #0", which is the first feature in the 8-bit vector, as shown in the referenced Figure.**

P26 L31: as the validation of this method is spatially limited, this sentence should be corrected.
**This has been corrected to estimating SIT with "comparable morphology"**

Technical corrections
P2 L9: wieth -> with
P7 L6: With -> with
P7 L7: need references.
P9 L11: I don't see ratio of keel depth and snow-sail height in the Table 1. **This is Hs/Hk**
P17 Table: replace "no constant" with "without constant".
P19 L16: replace "CNN" with ConvNet.
P22 Figure 10: Figure 10 never mentioned before.
**Thanks, these have been fixed.**

**Reviewer #2**

**Estimating Early-Winter Antarctic sea ice thickness from deformed ice morphology by Mei, J. M., et al.**

General Comments: GC1: I note that a dedicated "Methods" section is missing completely. It is Data followed by Results. There are places where this seems ok for the flow of the manuscript but there are other places, e.g. Section 3.2 where this seems not to be optimal. The deeper I stepped into this section the more confused I got. At a certain point I got lost with density values and with regressions with or without intercepts or additional constants. This section would perhaps benefit from a clear up-front explanation of what you did / how you derived coefficients / which density values you choose (and why) / how you derive effective density values (and why)? Such an improvement in structure of the paper would possibly also reduce its length a bit here and there.
**The paper has been restructured with the addition of a Methods section. The text and motivation for this analysis has in particular been improved by separation into Methods, Results, and Discussion sections, so that the basis for comparison with**

**ConvNet and previous results is more clear. We have also improved the discussion of regressions and derivation of density values (discussed more in response to the next comment)**

GC2: You treat the hydrostatic equation as a form of a linear fit. While one can see this as such a fit it would be very important to mention (even more) that the coefficients as you call them are based on density values and are computed based on physics. This is an important difference to the empirical linear fits used by Xie et al. or Ozsoy-Cicek et al. which are purely mathematical. To my opinion it would add to the understanding of your paper if you would clarify this even better at an appropriate position in your paper. I'd think that interpreting the CNN results into the direction that effective densities can be derived is very hypothetical - especially given the unknown (and non-existing) relationship between sea-ice and snow densities which are both involved. I note in this context that the issue of negative ice freeboards has neither been mentioned nor discussed. I guess it would not hurt to get back to it given the results published in Ozsoy-Cicek et al. (2013) and Yi et al. (2011).

**We interpret the hydrostatic equation as a linear fit solely to provide density estimates, which is only relevant for a dataset that has snow depth measurements. We now stress this more in the text. The ConvNet results do not derive any effective densities; we only suggest that the network may be accounting for different effective densities.**

**Similarly, the one-variable fits to surface elevation (F) can be interpreted as (given some snow/ice density) an average snow-to-ice ratio in the measured surface elevation. This again is not particularly prescriptive, as other datasets no doubt have different snow-ice ratios. However, when averaged over large enough areas, it is likely that the ice component in the measured surface elevation is low (and perhaps the snow = freeboard assumption is now reasonable in some, or many cases). This can be inferred from Xie/Ozsoy-Cicek's fits as their coefficients of 2-3 are equivalent to assuming F=D in our Eq. 1. We stress this more in the text that these no-constant fits are intended to check why our coefficients may be different to Xie/Ozsoy-Cicek's.**

**Also, we deliberately do not discuss negative freeboards in this case because in this dataset there are few negative freeboards sowhen averaged over 20m windows, our data have no negative freeboards (although at the 0.2m resolution there are some). and so they cannot be reasonably included in the article. Where they do exist, they are primarily on the flanks of ridges where their near-local effect will be negligible. We have added a sentence in the Data section to note that there were few negative freeboards.**

**Note, when applying either a linear fit or the ConvNet to surface topography data, we cannot know whether there are negative freeboards; as such these methods account for it only implicitly, with a linear fit effectively assuming that a similar percentage of freeboards will be negative. This may contribute to errors when trying to apply a specific linear fit to a new dataset. A ConvNet could conceivably do better here, in that significant negative freeboard is likely to matter most when there is deep snow, which might have recognizable surface morphology, although this is quite speculative. We have added a note about potential effects of negative freeboards in the discussion.**

GC3: I am missing the presentation / discussion of more results of the ConvNet approach. What a reader might have loved to see is profiles of sea-ice thickness computed from the draft-snow depth-surface elevation measurements (your benchmark)and of the sea-ice thickness estimated with your approach. Ideally you are able to show at least one representative profile of each PIP used here. That way one will get a better handle on the actually estimated sea-ice thickness distribution compared to the measured one - in addition to the histograms shown.

**This has been added to the manuscript. It is also shown here, using PIP8 as the test set. This plot requires considerable oversampling (here, it is oversampled at 4x, i.e. using a shift of 0.25 * window size of 20m), otherwise there are not enough points to make a useful visualization. The mean relative error of the ConvNet (trained on PIP 4, 7, 9) applied to this test set (PIP8) is 23%, vs. 31% for the linear fit (fitted to PIP 4, 7,**

**9) applied to this test set (PIP8). It shows that the ConvNet is better generalized to new datasets, and also shows the considerable biases that a linear fit can have when applied to a new dataset, presumably due to a varying snow/ice ratio. We have generated these for each floe, but choose to include one example in the manuscript as all are qualitatively similar.**

[Figure]

GC4: ICESat-2 is up since September last year. After having read the paper I am won-dering what the ultimate goal of your work is. Is it to create high-resolution validation data sets of the sea-ice thickness which are spatially distributed? Or is it to develop an algorithm which potentially could be applied to ICESat-2 data. For both cases, I believe the authors could stress the main motivation and future use of their work and product.

**This is a good point, and we have added to the text in the introduction to better state the goal of the paper. The long-term goal here is to improve on ice thickness algorithms for ICESat-2. However, our results are not directly transferable as ICESat-2 only maps a straight line (well, the 3 beams are not quite enough to form a 2D surface). As such, the ConvNet approach is not appropriate. While there are also non-convolutional deep neural networks that could work on such data, although one would need to test whether ICESat-2 can sufficiently capture morphological features that are related to ice thickness.**

**An alternative approach is then to use other more extensive 3D-datsets and use the technique to identify what morphological metrics are best predictors for a variety of types (which is much of the reason for our investigation of possible physical basis for the ConvNet features). For example, with IceBridge data one could use this method to predict radar snow depth. Alternatively, additional datasets (for example more coincident AUV and surface topography, or coincident scanning LIDAR and EM-bird observations) could be used to relate these identified metrics directly to ice thickness. Then it may be possible to relate ConvNet metrics that are good predictors of thickness to analytical metrics, then the results could be used to optimize algorithms for ICESat-2**

**Our goal in the paper was to demonstrate (1) to show that sea ice surface morphology contains information that can be related to ice thickness so that linear fits can be improved upon, (2) deep learning has the potential to use this information to provide "optimal" predictions of ice thickness, and (3) these deep learning techniques respond to features that are likely physically meaningful and hence there is scope to use this**

**physical information to provide better ice thickness predictions. We acknowledge that our particular deep learning architecture is not necessarily what would eventually be used in practice.**

**This was only briefly touched on in the Dicussion and Conclusion before, so we have expanded this discussion to better suggest a viable strategy.**

Page 2, Line 14: I suggest to cite the paper by Behrendt et al., 2013, Sea ice draft in the Weddell Sea, Earth System Science Data, 5, measured by upward looking sonars to underline that also this ULS data is a valuable source - even though you write "sporadic"
**Added, thanks.**

Page 3, Line 1-9: I guess it would not hurt to perhaps again refer back to Kern and Spreen (2015) who dedicated some work on the uncertainty analysis for ICESat sea-ice thickness retrieval and to Kern et al., 2016, Antarctic sea-ice thickness retrieval from ICESat: Inter-comparison of different approaches, Remote Sensing, 8(7), who inter-compared a number of sea-ice thickness retrieval approaches for the Antarctic and on which results the Li et al. (2018) paper cited is based upon. I am tempted to say that the fits used by Li et al. (2018) are based on the work of Ozsoy-Cicek et al. solely and not on the work of Xie et al. (2011).
**Thank you for the correction on Li (2018). This and the additional references have been added to the introduction**

Line 14: "do not yet understand the distribution" –> I am inclined to say that we do understand the physical mechanisms forcing the snow depth distribution around ridges very well. What we cannot yet do is, however, to measure this distribution accurately over an large enough area.
**Changed to "we do not yet know the statistical distribution"**

Line 10-19: In this paragraph, the work of Weissling and Ackley, 2011, Antarctic sea-ice altimetry: scale and resolution effects on derived ice thickness distribution, Ann. Glaciol. 52(57) might fit as well.
**Thanks, added.**

Lines 16/18: This is perhaps a good place to refer to the work of Hutchings et al., 2015, Comparing methods of measuring sea ice density in the East Antarctic, Ann. Glaciol., 56(69)
**Thanks, added.**

Lines 20-22: I agree that the unknown snow depth is one factor here. But isn't the fact that we don't know the keel morphology and distribution relative to what we see from above with a LIDAR contributing much more to a potential bias in estimated sea-ice thickness?
**Yes, this is a good point, and we have clarified this in the text.**

Line 23-27: As far as I know, Kern and Spreen (2015) focused quite a bit on ICESat and the uncertainties involved. I doubt, however, that this is the correct citation for the AMSR-E snow depth bias issue. I'd say the first to report this issue were Worby et al., 2008, Evaluation of AMSR-E snow depth product over East Antarctic sea ice using in situ measurements and aerial photography, J. Geophys. Res., 113. Their work was followed later by Ozsoy-Cicek et al., 2011, Intercomparison of Antarctic sea ice types from visual ship. RADARSAT-1 SAR, Envisat ASAR, QuikSCAT, and AMSR-E satellite observations in the Bellingshausen Sea, Ann. Glaciol., 52(57) or Kern et al., 2011, An intercomparison between AMSR-E snow depth and satellite C- and Ku-band radar backscatter data for Antarctic sea ice, Ann. Glaciol. 52(57). In the same Ann. Glaciol. volume you also find the paper by Markus et al., 2011, Freeboard, snow depth and sea-ice roughness in East Antarctica from in situ and multiple satellite data, Ann. Glaciol., 52(57). Another paper about the deficiencies of the AMSR-E snow depth product could be this one: Kern and Ozsoy-Cicek, 2016, Satellite Remote Sensing of snow depth on

Antarctic sea ice: An inter-comparison of two empirical approaches, Remote Sensing, 8(6).
**Thank you for the detailed notes; text and citation has been amended.**

Line 33: "fewer such datasets exist" –> This applies to the Antarctic and I would mention this accordingly. In the Arctic there are way more draft measurements available and these have actually been used to develop draft-based sea-ice thickness estimation tools.
**Fixed.**

Figure 1: I love this figure. It could be even a tiny bit more realistic if the ice floe or sheet would not be continuous in the ridge / keel area.
**This is fixed.**

Page 4: Lines 3-12: I am wondering whether in the context of this discussion the work of Goebell, 2011, Comparison of coincident snow-freeboard and sea ice thickness profiles derived from helicopter-borne laser altimetry and electromagnetic induction sounding, J. Geophys. Res., 116 should be mentioned as well?
**Thank you for this reference, though it is added a bit later for discussing the coefficients for T vs F in later sections.**

Page 6: I suggest to mention / give answers to the following questions here: - Water depth in which the AUV was operating - How many AUV scans per "cake" were stitched together? - If multiples scans: Were all scans carried out into the same direction? Or parallel to each other in opposite directions? X-ing? - Did I understand correctly that per "cake" 4 surface elevation scans were carried out, each from one side of a 100 m x 100 m grid? Or did you actually fly over the area? - I assume the snow depth measurements were the last measurements carried out –> although it is logical it is worth to mention this. - It is not entirely clear how the about 2000 measurements per "cake" are distributed across the "cake" area and how this was technically realized. I assume that the measurements were carried out along parallel transects across the cake with a fixed transect-to-transect distance and that only the sampling along each single transect varies between 5 m and 0.1 m. - How is the reference sea-surface height computed and how accurate is it?
**Added to text.**
**The AUV survey was done following Williams et al (2014), at a depth of 15-20m in a lawnmower pattern (equally spaced passes under the ice in alternating directions). Adjacent passes were spaced to provide approximately 50% overlap in consecutive swaths, with at least one pass across the grid in the transverse direction to allow corrections for sonar orientation in the stitching together of the final sonar map.**
**The snow depth was indeed done last. Your description of the snow sampling is mostly correct, the sampling along a transect is not purely 1D, as we could not necessarily walk over a ridge, and instead would walk around it, sampling the snow distribution at high resolution.**

Line 29: "The ice thickness can ..." –> so there were no drillings?
**There was one drilling line done per floe with 50 points at 2m resolution along the edge of the lidar scan. This was used for calibration purposes using the level ice. This is described in the Data section. An example of the comparison is shown below, but is not included in the manuscript for brevity. The freeboard measurements (middle panel) have poor agreement due to sampling bias (often, when drilling near a ridge, the most accessible point to drill is the lowest elevation). The error bars here refer to the max/min MagnaProbe/Lidar/AUV measurement in a +-1m range. The drilled draft measurements do not necessarily match the AUV measurements in ridges because there may be biases in drilling through thick ice, slight differences in location when thickness variability is extreme, and may tend to miss deeper loose blocks. The AUV will tend to detect the full draft, and samples at the sonar footprint resolution (as opposed to at a 2 inch drill hole point)**

[Figure]

Figure 5: Readability of this figure would improve with an increase of its size.
**Fixed**

Page 11:
Line 18: "very similar value of 1.3" –> This very similar value needs two standard deviations (2 time 0.1) to include that 1.5 values from other studies. Perhaps "similar" would do it?
**Fixed**

Figure 6: - See comment to Figure 5. - In the caption I would call the black line dashed rather than dotted.
**Enlarged and fixed.**

Page 12:
Line 11: "The lidar and AUV data were corrected by …" –> I don't understand what needs to be corrected here. Is there a way you specify better what you did? Why is this a "correction"?
**The correction is a simple offset to the entire lidar or AUV survey that is applied to account for the uncertainty in the AUV trim (which may cause slight depth offsets for different surveys) and the depth sensor, and lidar referencing to the sea surface (linked to your earlier comment). This has been clarified in the Data section.**

Page 13: Line 12: "snow = surface elevation assumption" –> I recommend to mention that this is a strong assumption, that it applies to thin, perhaps medium thick first-year ice only (possibly only to the kind grown under quiescent conditions, i.e. not originating  from the pancake-ice cycle), that it requires a certain snow load to be present, and that such an assumption can only be made if one is not interested in a really exact sea-ice thickness estimate.

**This has been clarified so that it is clear this is a lower bound, and that in our case it does occur for the thinner, level ice.**

**However, we note that this is quite often close to being true, at least for regional means such that the large-scale error is small. Based on our own observations in the field over many cruises, it is broadly true for much first-year ice, and even can be reasonable for very thick ice where positive freeboard and negative freeboards cancel. While this is often reasonable on a regional scale, we agree it is not a good assumption at smaller scales, and presumably what creates much of the scatter in linear fits. Incidentally, this also suggests why using surface morphology may help improve predictions – even where a linear fit is accurate in the mean, it cannot capture this variability, while surface morphology may be suggestive of variations in the snow/ice freeboard ratio.**

Line 17: "All our coefficients" –> Would you mind to refer to the place where you already mentioned these coefficients?
**Added**

Line 21: "2.2-3.1 in Ozsoy-Cicek et al (2013)" –> I tried to figure out how you ended up with this range and potentially misunderstood something. If I check that paper, then - in Figure 5, which is possibly the one you got these numbers from, - I find regression lines with a considerable intercept between ~10 cm and ~30 cm, depending on the region, paired with this range in factor of F of 2.2 - 3.1. But these values are valid for positive ice freeboards only. When taking all ice freeboards into account, then the black lines (and numbers) in that Figure 5 apply. In addition to that: I could make sense to focus only on the Ross Sea results from that paper?
**We used the positive freeboard regressions because all our 20m-averaged freeboards have no negative freeboards (although there are individual negative freeboard values at 0.2m resolution).  If we take the 'all freeboards' coefficients, the range is 2.4-3.5, which does not affect our analysis. However, if we just take the Ross Sea coefficients then the range is 2.4-3.1, which again does not change the analysis. So we will just use 2.4-3.5 as our analysis would apply to both cases.**

Lines 30-32: "This means that assuming ..." –> So what you state here basically is, that the linear regression approaches developed by Xie et al. and Ozsoy-Cicek et al. are of limited value? If so you could mention this and also refer to Kern et al. (2016) in Remote Sensing, where it is layed out that the linear regression approaches fail to provide a meaningful circum-Antarctic sea-ice thickness distribution.
**Added. However, the approach of Xie et al and Ozsoy-Cicek et al may be reasonable at larger scales, and this is described in the text. We now state that such relationships should be used with caution.**

Figure 7 - Some data points are annotated "snowy" –> I did not find an explanation of what this is in the text or in the caption. Where is the distinction between "snowy" and "ridged"? - caption: The ice density value given in line 4 of the caption differs from the one given in the text on page 13, line 25.
**Density value changed. 'Snowy' surfaces are manually classified as those that have snow features (likely originating at the ridge, but the ridge is not in the window).  The classification is purely meant for the feature analysis in the Discussion.  This is described in the text, but now we also added "manually classified" to the caption.**

Page 14: Line 3: "T = 2.45F + 0.21" –> is modified from Ozsoy-Cicek et al., now using unit meters instead of centimetres, correct? "for a winter Ross Sea" –> according to Ozsoy-Cicek et al. (2013) this is data from just one cruise in Sep./Oct. = much later in the season than PIPERS. In that sense your statement in Line 6 "same region/season" should perhaps be changed? Also the spatial overlap (see Ozsoy-Cicek et al., 2013, Figure 1) is quite small.
**Yes we converted the equation to meters. Merging with your below comment for lines 15-17, we have removed "same season/region", and now point out that the proportion of deformed ice is varying and perhaps causes linear fits to not generalize well.**

Line 4: So your intercept is -0.73 meters or -73 cm? That is quite large.
**Yes; we discuss the reasons for this in the text.**

Line 7: "nonzero freeboard" –> "nonzero ice freeboard"
**OK, changed**

Lines 15-17: Yes, I agree with your interpretation. However, it might make sense to also mention that the Ross Sea data used in Ozsoy-Cicek et al. (2013) was from a different part of the Ross Sea and from a different season and to my opinion indeed exhibits a totally different characteristics than the PIPERS data set collected 3-4 months earlier.
**Yes, see 3 comments before this for summarized changes.**

Page 15: Line 6: "additive constant" –> I don't understand what you mean by this. Did you add an intercept?
**Yes, changed this to "we fit a linear regression both with and without a constant term" - we don't want to use 'intercept' as this only has meaning for a one-variable fit.**

Lines 7 & 8: Isn't it surprizing that the coefficient for F fitted over all four PIPs of 10.4 is so close at the upper range of 10.6 for individual PIP fitting? Also: The range for the coefficient for D of the individual PIPs does not include the value found over all four PIPs. Is this logical?
**The answer to both of these questions is that the multilinear fit fits both variables simultaneously, and so the fit for all floes combined is not a weighted average of each individual fit (your intuition would be correct for a one-variable fit – and indeed it is, the F-only, no intercept fits have coefficients of 6.5, 6.4, 4.8, 4.1 and the overall fit is 5.8)**

Lines 9-12: - Your measured snow densities are considerably lower than those given by Sturm et al. (1998). Could it be that the latter were obtained in late winter / spring? - While I understand the concept behind the effective sea-ice density (voids filled with water included in the density estimate) I have problems to understand the concept of an effective snow density. What is this? In this context, I find your effective snow density value to be quite high. - I guess it would be good to learn how you ended up with the density values reported in Line 10. When I tried to insert your range for the factor for F (7.9 to 10.6) into an equation where D is zero, then I end up with densities between 897.9 and 931.0 kg/mˆ3. But of course, without further information from your side I cannot reproduce your numbers. - I find it quite surprizing that the standard error for the effective sea-ice density is so low compared to that of the effective snow density.
**Sturm includes 2 Ross Sea cruises, one in May-July 1995 and one in Aug-Sept 1995, with snow densities of 350 and 390 kg/m3 respectively, although the May-July cruise would have somewhat older snow than in our case (due to somewhat earlier dates and a later freeze-up for PIPERS)The snow does not have an "effective density" and this has been corrected in the text.**
 **The standard errors are computed using the standard error of the linear regression and propagating them. This should not really be interpreted as an uncertainty value for the sea ice density, as it just means the multilinear fit has a (relatively) low error for the F coefficient (sea ice density variability can still contribute to displacement of any given point from the fit line) This has been clarified in the text.**

You use a water density value which differs from those given at the beginning of this paragraph. Why? Where does this value originate from?
**The water density of 1028 comes from CTD casts from PIPERS, whereas 1027 comes from Worby (2011). The difference is minor, but have added that this came from onboard measurements.**

Lines 15-18: Please check these sentences. There is some repetition first and then something is missing.
**Fixed**

Lines 20-22: "For example, ..." –> Just to understand this: What you write here in the text is the comparison between using coefficients of ONE of the PIPs to estimate sea-ice thickness in another PIP while in Table 3 you show the comparison between using a joint coeffient of THREE PIPs to estimate sea-ice thickness in the remaining PIP. I just got confused a bit about why you write different things in the text than you actually show in Table 3 (and refer to in the subsequent sentence).
**You are right. We had listed the results of using each ONE of the PIPs to show that the average error was not dominated by one particularly bad one, but you are right, it is better just to show the average of the THREE PIP fit applied to the fourth PIP.**

Table 2: - "no int." means what? - Are you sure the AIC is a monotonic function even on the negative value range? I am just wondering whether the "smallest" AIC criterion does not need to be applied to absolute values? Could it be that these negative values have no proper meaning in case that the correlations are so low? - The subscript "adj" stands for "adjunct"? "adjusted"? If the latter adjusted to what? - What is the unit of the constant? It seems to be in meters?

**No int. means the forced fit through the origin with no intercept. This has been added to the caption.**

**The AIC is not dependent on the absolute measurement, and rather the difference between AIC values can be interpreted as a relative likelihood, e.g. if two models have AIC values A and B, with A<B, then the second model is exp(A-B) times as likely to minimize the information loss.**

**Adj stands for adjusted, and it is adjusted to account for varying sample sizes. This is mentioned in the caption.**

**The constant is in meters and this has been added to the column.**

Page 17: Table 3, caption, last line: "zero freeboard = zero thickness condition"? Do you refer to ice freeboard here? Do you perhaps mean "zero snow depth"?
**This has been deleted from the caption, as we now include the fit with constant term.**

Line 1: This equation is something you could use in a "Methods" section (should you include one) to tackle my general comment GC1. I note however, that seemingly with this equation one can explain only parts of the entries in Table 2; the c3 times sigma part is not represented in Table 2.

**The fit with sigma is only mentioned to show that it does not improve the fit (likely because sigma is itself highly correlated with mean surface elevation). It is not an important result by itself, and also would not fit into Table 2. The equation will be moved to the new Methods section.**

Lines 5-9: It is still not clear to me how you discriminate between "snowy" and "level". Please add.
**This is manually done based on whether the majority of the image was level or contained a visible snow feature in the lidar window. We acknowledge that this classification can be arbitrary, and use this method only to show that different surface types should be treated differently, but a manual classification does not help much: this motivates the use of a deep neural network in the next section.**

Page 18: Lines 3-6: I can in principle follow your argumentation that zero surface elevation (= zero snow depth) means zero sea-ice thickness. I would sign this if we consider larger scales. But on the scales investigated with the PIPs this is not necessarily true because under cold conditions and hence impermeable sea ice there will be many places with a negative ice freeboard. Even if we assume for simplicity that most of these will have a snow cover and hence potentially have a non-zero surface elevation, it is still likely that especially in the vicinity of ridges and/or where the ice is under lateral stress - at the scales of your measurements - you will have surface elevations
close to zero or even negative ones paired with a non-zero sea-ice thickness. - This paragraph is again a good place to comment and/or underline the difference between the physically based coefficients used by Zwally et al. and similar papers and the empirically based coefficients used by Xie et al and similar papers. One could argue that the physically-based coefficients are more dependent on the validity of the hydrostatic assumption while the empirically based ones are not ... but I am not sure this holds.
**On the scale of the actual linear regressions (20m), there are no negative (mean) surface elevations. However, we have decided to scrap the requirement of zero S.E. = zero thickness, as our ConvNet performs better than a linear fit with constant anyway.**

**As stated above, the use of no constant is now only used to compare to the theoretical fit and estimate densities.**

Page 19: Line 13/14: "20% of the data ..." –> this refers to the randomly selected data? If so, please stress so in the text.
**This has been clarified.**

Line 13 vs. Line 16 and remainder of the text: Please check your usage of "floe". From the text until here I got the impression that the PIPs are subsets of one floe. Here I get the impression that PIPs comprise several floes out of which a few are selected. Please clarify your terminology here.
**Each PIP is sampled from a different floe, but is nevertheless a subset of that floe (i.e. only a portion of the floe is sampled. This has been clarified in the text.**

Line 20: You state PIP8 here but in Figure 9 it seems you refer to PIP9. Please check.
**Fig. 9 should read PIP8, thanks for noticing this.**

Line 24: "epoch 881" –> does it make sense to refer to the Appendix here? Otherwise this information is perhaps a bit out of context.
**Ok, this has been moved. Note that most of the appendix is now in the Methods section, in response to other reviewers and to not repeat information from the Methods.**

Line 31: "are all negative" –> except for level ice.
**Fixed**

Line 33-35 and beyond: I doubt that this comparison should be presented as is. Aren't these data sets quite different? I wrote about the sub-set of data for the Ross Sea used in Ozsoy-Cicek et al. (2013) already. In Li et al. (2018), the data basis is ICESat footprint-scale estimates of the freeboard - hence we talk about one value for one footprint of which we do not know how well it covers how many different surfaces. The data set used in Ozsoy-Cicek et al. (2013) is at least based on multiple measurements conducted on one or more transects across a single floe.
**This has been amended to compare against the profile mean RMS error of 11-15 cm from Ozsoy-Cicek et al (2013) Table 7 against our validation error. The training error, which is equivalent to a fit error, is not as good of a comparison because a ConvNet can/will overfit with an artificially low training error; model selection is done by choosing the best validation error, which is kept separate (but is similar to) the training set in order to try and reduce overfitting.**

Figure 9: The description / caption of the figure needs to be improved. - Please annotate the images with a), b), c) ... - What is the value behind showing a continuous fit in addition to the bars? It extrapolates the bars towards non-existing data values. - What are the bin-sizes used? Are these the same in all three histograms shown? Do they always have the same borders (i.e. minimum and maximum value included into the count of a respectiv bar)? - The peak counts are obscured by the legend. This needs to be changed. - I suggest to add in each row which PIPs are used for what. - I suggest to stress in the caption that the last row shows a different range of thickness values. - The caption in line 3 says PIP7-9 but in line 2 it is PIP4, 7 and 8. What is correct? - The caption in line 4 says PIP9. True? - What is the unit at the y-axis in the histograms? - The mixed colors in the histograms originating from overlapping bars of different data sets are not easy to interpret. Perhaps you could either add these in the annotation (which is possible if you increase the size of Figure 9) or find a different way to show the counts of the different data sets. One way to do this would be that you use substantially narrower bars which you do not let overlap each other and center these at a specific thickness; then in the caption you might need to state that you display three bars centered at a specific thickness, separated horizontally for better visibility. That way the real differences in the distributions would become more clear.

**This figure has been redone with just the outline of the histogram, binned at 0.4m, with additional labels added. The caption has been fixed, thanks for noting this.**

**The bin sizes were chosen to have equal numbers of bins as opposed to a constant bin size; this has been changed to constant bins of 0.4m.**

Page 20: Lines 2-4: Perhaps you could put these numbers in context with the number of data points used to get these uncertainty estimates? I guess, in case of Ozsoy-Cicek et al. (2013) we are talking about 23 floes with an actually unknown number of measurements per transect. About how many measurements are we talking in your case?
**We have 4 floes, but each floe has many measurements as the lidar/AUV data can be binned at varying resolutions. So the REM (relative error of the mean) is comparing floe mean with floe mean; although as you point out the floes may have different numbers of measurements. We have added a note that our test error is essentially taking 3 floes and applying the fit to a fourth floe, vs. fitting to 23 floes. It is probably easier to get a good fit with fewer floes, but we also expect poorer generalization with fewer floes, so we can reasonably infer that our fit is better generalized than a linear fit.**

Page 21: Line 14: I suggest to remove the "see" –> at least I cannot see these results.
**OK**

Line 29: "isostatic assumption may no longer be valid" –> may be so. What do we know about spatial scales over which the isostatic assumption is valid? No too much I'd say - particularly for ridged ice. Perhaps this sentence could be deleted.
**Deleted. In fact, we realized that the ConvNet does not require assuming isostacy.**

Figure 10: - this figure belongs to section 4 and should be located within section 4 not before it. - Why do we have 3x3 imagettes for layer 1 but 16 for layer 3? - The size of 4 m and 8.8 m given in the caption, do these refer to the pixel size in these imagettes or to the imagette size itself? It seems as if the pixels in layer 1 are indeed smaller than in layer 3. - Instead of "as the lidar" you might want to write "as the surface elevation" - Is it in this context correct to assume that layer 3 has the unit meters while layer 1 is unitless? - What do the bright and dark pixels in layer 1 mean?
**Moved to appropriate section. The pixels corresponding to the meters of the layers have been added to the caption. Due to the stride, each subsequent layer essentially halves in resolution. Darker colors indicate higher weights, though the actual weight values are not important. The reference to "surface elevation" has been fixed. All the layers are unitless as the weights are just a numerical weight value. The direction of the colorbar doesn't actually matter as the difference would just be a negative sign, easily accounted for in any of the subsequent hidden layers. This has been clarified in the text.**

Page 22: Line 17 through Page 23: Line 7 and Figure 11: - Please provide a), b), ... in Figure 11; it aids referring to the images. - I suggest to mention Figure 11 before Figure 12. - You refer to Figure 11 in Line 5 but should perhaps also do it in Line 3 (strong correlation for feature #0) and again in Line 6. - I have difficulties to understand the continued mentioning of "effective densities". I doubt that with the CNN you can (and should) derive any conclusions about the effective density - especially because the densities for sea ice and snow do not necessarily co-vary. This brings be back to GC2.
**Figure 11 and 12 have been switched. The ConvNet cannot give any conclusions about why it has learned its prediction; we simply try to give physically plausible explanations without asserting that these are true. Our goal is to show that this method can work in general for other datasets, and why we may expect this to be the case. We will attempt to better stress how speculative our discussion is.**

Page 23: Line 10: "snowy surfaces" –> which still need to be defined in comparison to

"level surfaces".

**Yes, this has been clarified in the text.**

Page 24: Line 4: "ridged and level surfaces are clearly distinguishable" –> I don't agree when I look at Figure 13 - unless I have perhaps misunderstood what the used tool is able to show. But my interpretation of this figure is that level, ridged and snowy symbols overlap well.

**Ridge, snowy and level overlap somewhat, but Ridge and Level are more distinct with less of an overlap. This suggests their features are differently analyzed by the ConvNet.**

Page 25: Lines 9/10: That prediction of snow depth from lidar input is possible as also been shown by Ozsoy-Cicek et al. (2013) and Kern and Ozsoy-Cicek (2016).

**OK, added**

Lines 11-13: I guess these two sentences could be deleted.

**Removed**

Page 26:

Lines 1-8: I am not a fan of these attempts to try to relate CNN features to (effective) snow density variations which may or may not be realistic and physically meaningfully linked to input parameters. To my opinion, this really requires a careful analysis and description of how the CNN "learns" from the input data and whether there is (within the CNN) a link to physics - which I doubt is the case.

**We agree that the speculation that these features may be linked to snow density variations is highly speculative. Because the weights of these features (5, 7) are so small, we decided it is not that important and we have removed this from the text.**

Lines 18: "thickness of a new dataset" –> you seem to have applied your approach to a different PIPERS data set. It might be really beneficial to show this example in the paper and not to just mention it. Particularly because you come back to this in your conclusions (Line 33, "unseen floe").

**This was badly worded. By new/unseen floe, we mean a dataset on which the net is not trained (i.e. the test floe). We have changed these to "test dataset".**

Page 27: Line 5: "it can account for a varying ice/snow density" –> I'd say that this is a hypothesis. It may be that the ConvNet is able to account for the different densities and perhaps even provide additional information about these - but the evaluation of whether this is the case and/or whether this is at all meaningful physically based on deep learning is not known and might not be over-stressed here.

**We have softened this as a "possible" strength.**

Lines 12-15: "Our error ..." I suggest to not overstress these inter-comparisons be-cause these are based on completely different data sets and scales. After all, a real quality measure of your method will be its application to ICESat-2 data which should be the overall goal here - as is finally mentioned in the last paragraph.

**We will keep the reference to the survey-wide mean RMSE from Ozsoy-Cicek 2013 because as explained in the results and discussion, we feel this is a reasonable comparison, but we temper the statement by saying these are different datasets. We will delete the comparison to Kern 2015 and satellite based estimates.**

References: You need to go through the references list and complete it with respect to page numbers and journal volume and issue numbers. Also doi's are generally missing. Some journal abbreviations are not in place.

**OK - done**

Page 28: Line 31: I am not sure but I guess Figures in the Appendix need to be named differently to the main text. See the instructions for authors.

**Thanks for bringing this to our attention.**

Page 29: Line 2: Would you mind commenting on the layer sizes being first 4 m, then 8.4 m and subsequently 8.8 m? Do these "strange" values have to do with the pixel size of 0.2 m?
**This is discussed in an earlier caption, but it is simply because of the stride of 2 halving the resolution each layer (so 0.2m, then 0.4m, then 0.8m), with window sizes of 20, 21 and 11 pixels. This has been clarified.**

Figure 15: - What is a "training loss"? - It appears that after epoch ~550 there is a small jump in validation and training error from a certain level before that epoch to a certain, lower level afterwards. Any explanation to this? - What explains the sudden increases in the training error from a low background of ~15-16% MRE to the level of the validation error? It seems as if the result of the ConvNet even after that many epochs is still not stable?
**The training loss is just the loss function (mean squared error) of the training set. This has been clarified in the caption.**
**The jump may be due  the method being stochastic and this accounts for the error jumping around. Also, because we are optimizing mean squared error, this is correlated to but not exactly equivalent to optimizing the MRE (which is closer to optimizing mean absolute error). As the method is also stochastic, we could possibly get a smoother curve with more epochs and a smaller time step, but this increases training time. Again, the point here is to show that this method is effective for lidar datasets in general and not to propose that our architecture is the best possible one.**

Typos:
Please replace "e.g" by "e.g." (a few incidences) Please check usage of "climatology" and replace all incidences in the paper by a more appropriate term.
Page 2: Line 9: witeh –> with, interrannual –> interannual
Page 9: Line 26/27: "beyond beyond" –> "beyond"
Page 15: Line 11: "which" –> "who"
Page 21: Line 28: "slighly" –> "slightly"
Page 28: Line 23: "assigining" –> "assigning"
Figure 14, caption, line 3: "optimzer" –> "optimizer"
**Thanks for this.**

**REVEIWER #3**

Dear TC editor and authors of the manuscript TC-2019-140,
The topic of the manuscript is interesting and the content useful for sea ice research. A neural network has been applied to sea ice thickness (SIT) estimation from lidar surface elevation. The introduction section is quite comprehensive. It is mentioned that it may be possible that the introduced method may be used to improve SIT estimation by lower resolution / larger footprint laser instruments (ICESat-2). The results have mostly been presented nicely and comprehensively.

The first referee already submitted quite comprehensive comments on the manuscript, and I'll just try to complement his comments. I agree with him that a major review is still required before publication

Here are some comments trying to improve the manuscript:
General comments:
1) I agree with the reviewer 1 that more results of the DCNN approach could be included.
**We have added an additional figure into the DCNN Results section showing the spatial distribution**

2) Also a more detailed technical description of the applied methods (DCNN) would be preferable, as already suggested by referee 1. This could be a Section of its own (not an Appendix). Also include the information of numbers of DCNN neurons used at each

layer and how these numbers were selected.
**We have added a Methods section with most of the material that was in the appendix.**

3) Regarding e.g. icesat-2 data, it would be nice to have some experiments or at least approximation related to the effect of resolution to SIT estimation using the proposed method.
**Unfortunately, ICESat-2 data is linear and not suitable as an input for our ConvNet, although see the response to reviewer #2 above regarding how these results provide a demonstration of deep learning techniques and a possible path to an improved ICESat-2 algorithm. We have discussed a halving of the resolution and its effect on the accuracy in the Results. We cannot reduce the resolution too much as each lidar scan is only 100m x 100m, which limits how large our window can be. We are now exploring this with ICEBridge data, but this will be a subsequent paper.**

More detailed comments:
S. Introduction P5, L17-18 "... detailed snow depth measurement": Also include al-
ready here by which method the snow depth measurements were made (not in detail).
**Added "manually-probed"**

S. Data P6, L12 and L22: instruments are named, also include references to their technical specs, and also shortly write on the principle of the snow measuring device.
**Added.**

S. Data P 6-7: Division of the data sets used into training and test data sets (possibly also validation data set) could be clearly described in the data section already. Were the data sets the same for all the performed experiments? This seems to be described later in the deep learning section for the DCNN.
**We have added a Methods section which describes this training/validation procedure and the test data set**

S 3.1.1 P9, L18-19: Rather say "...Thickness of the level ice (L) forming a sail and its sail height (S)..."
**Fixed, thanks.**

S 3.1.1 P10, L8 "...for estimating sea ice thickness,..." -> "...for estimating sea ice thick-
ness T,...". Possibly You could use SIT for sea ice thickness throughout the manuscript?
**We have replaced sea ice thickness with SIT.**

3.1.2 title could be "...mean sea ice thickness..." or "...mean ice thickness..." or even "mean SIT".
**Changed to SIT**

P10, Fig. 5. Make the figure larger, difficult to read in the printed version. Its width could e.g. be approximately the column width.
**OK – done.**

P11 S 3.1.2 L22: Describe the use of semivariogram in more detail. Did You make any experiments by varying the window size also?
**We used the semivariogram to identify the optimal window size. We did try a half-sized window, as described in the text, but with somewhat worse results, likely because the windows fail to capture surface features.**

P12 Fig. 6: Same thing as for Fig. 5, make larger.
**OK – done**

P14 Fig. 7: Same thing as for Fig. 5, make larger.
**OK – done**

P18 Fig. 8: Make the figure larger or make the box frames wider for better visibility. Include a legend describing the classes instead of writing it in the caption.
**OK, done.**

S 3.3, P 19 L5: The best-performing linear regression result has been given here for comparison. Have You any idea, could better results have been achieved by using a nonlinear approach with the same inputs, e.g. a multilayer perceptron neural network with the same inputs (plus an additional constant/intercept input)? Or are the dependencies really linear?
**MLPs actually are less effective but more complicated than ConvNets due to their fully-connected style. This means the total number of parameters quickly becomes very high. We also use a nonlinear activation function in our ConvNet, as the dependencies are nonlinear, as you note. Moreover, we want a convolutional approach precisely because we believe the spatial information in the lidar 'image' is important and necessary for accurate SIT estimation.**

P19 L12-13: "20m x20m windows", also give the window size in pixels here.
Did You study the effect of the resolution to the result by using down-sampled data? Any idea, how would this possibly affect the estimation result? Possibly You could then get average SIT over a larger area? This could give an idea of the applicability of the method to coarser resolution data.
**Yes, we have tried to halve the resolution. This is mentioned in the text (was formerly in the appendix) and results given (a modest degradation in performance). As mentioned above, we cannot keep halving the resolution as then our dataset becomes too small to do any meaningful convolution. We are attempting to do this in future studies using Operation IceBridge data.**

Figs. 9, 11,12,1314 and 15: make bigger for better readability in the printed version.
**Ok – done.**

App. A: did You also vary the number of neurons at each level and how did this affect to the results? How were these parameters selected? Does there exist any "rules of thumb" for selecting the parameters (e.g. numbers of neurons) for DCNN's as there exist for Multilayer Perceptrons (as a function of the number of inputs and outputs)
**We varied the number of filters at each layer and if there were too few, then the results were worse. There are no rules of thumb, other than to double the number of filters in each layer if the stride is 2 (as the dimensionality of the data is halved), which we did. Again, we stress that our architecture could be fine-tuned to improve accuracy even more, and we simply aim to show that this method can be applied to improve SIT estimates.**

A: Also include execution times for the training and SIT estimation in the used hardware.
**Added.**

And yet one interesting aspect: As a researcher of microwave and optical EO imagery (over sea ice) I am also interested in possibilities of utilizing the existing imaging devices for SIT estimation. Typical high-resolution (HR) sensors covering a wide spatial area, such as HR SAR or optical/IR sensors, measure only the 2-D sea ice surface, not the elevation directly. However, it is possible to locate ice ridges and even estimate their sail width in HR EO imagery. There is some literature (e.g. Timco & Burden, 1997) relating the ridge parameters to each. However, I have not seen any good reference relating sail width (Ws) to sail height (Hs). This kind of relationship would be very useful for better estimating ice thickness from 2-D HR EO data. Could the authors comment on this topic i.e. how (well) the morphology could be derived/estimated from the available 2-D EO data/imagery and whether this relation could be utilized in SIT estimation? Possibly a deep neural network could be used after deriving some ridge parameters from 2-D HR sea ice data form SAR/optical/IR, or even just training

a DCNN with the data directly. This would naturally require a good data set with a large number of (nearly) simultaneous SIT measurements (possibly made by another validated remote sensing method, such as laser scanning).]

**We agree convnets might have utility to determine potential relationships with other metrics and sea ice thickness. For example, it is reasonable to think that spatial variability in SAR signatures might be correlated with deformed ice percentage, ice types, etc, which are likely correlated with ice thickness. Alternatively, HR imagery may show features that are indicative of snow dunes or ridges. However, at present we have little basis to expect that any such relationships might be sufficiently reliable, and without carrying out such analysis, we feel this is too speculative to comment on. To take the example suggested by the reviewer, our experience with analysis of ridge morphology in the Antarctic (from our data and Icebridge) we have not seen any suggestion of a relationship between the sail height and width. Ridges identified in imagery may vary from well-behaved triangular ridges which may exhibit some relationship, to rubble fields, which likely do not. Note also that in our case, it appears that the CNN appears to heavily use the freeboard; without any freeboard information, we do not expect a CNN to be very accurate in predicting thickness. That said, we would agree that a CNN would likely be effective in identifying ice types (in analogy to how a trained analyst does this).**

**Since any suggestions here would be very speculative, we prefer to not discuss here, although we have added additional discussion in the conclusions relevant to ICESat-2, as requested by the other reviewers.**